



# Projected changes in droughts and extreme droughts in Great Britain strongly influenced by the choice of drought index

Nele Reyniers[1], Timothy J Osborn[1,2], Nans Addor[3], and Geoff Darch[4]

[1]Climatic Research Unit, School of Environmental Sciences, University of East Anglia, Norwich, United Kingdom
[2]Water Security Research Centre, University of East Anglia, Norwich, United Kingdom
[3]Geography, College of Life and Environmental Sciences, University of Exeter, Exeter, United Kingdom
[4]Anglian Water Ltd., Huntingdon, United Kingdom

**Correspondence:** Nele Reyniers (N.Reyniers@uea.ac.uk)

**Abstract.** Droughts cause enormous ecological, economical and societal damage, and are already undergoing changes due to anthropogenic climate change. Understanding, anticipating and communicating these changes is essential to a wide range of stakeholders. In this study, the projected impacts of climate change on future atmospheric droughts in Great Britain were assessed for two warming levels (2 °C and 4 °C above pre-industrial levels) using the UKCP18 regional climate projections.

As projected changes can be very sensitive to the choice of drought index, two indices were compared: the Standardized Precipitation Index (SPI), and the Standardized Precipitation Evapotranspiration Index (SPEI, which unlike the SPI, accounts for increasing potential evapotranspiration). The SPI and SPEI were used to quantify drought frequency, extent and duration of all droughts and of only extreme droughts. To provide context, aridity and seasonal precipitation and potential evapotranspiration changes were also assessed, as well as seasonal contributions to dryness at a yearly time scale. The UKCP18 regional simula-

tions project (strongly) increasing drought frequency and extent due to climate change based on the SP(E)I almost everywhere in Great Britain. Importantly, the relative increase in frequency and extent is much more pronounced for extreme droughts than for more moderate droughts. Increasing longer-term dry conditions can be attributed mostly to more frequent dry and extremely dry summers, for which normal to wet winters are decreasingly able to compensate (even where winters are projected to become wetter). In general, using the SPEI results in far greater increases in drought frequency and extent than using the

SPI. These differences are so substantive that at +2 °C the SPEI6-based projected changes reach a similar magnitude to the SPI6-based changes at +4 °C. Finally, projected changes in the distribution of drought durations depend on the drought index, region and warming level. These results illustrate that the choice of atmospheric drought index can have a decisive influence on changes in projected drought characteristics, and therefore users of these indices should be aware of the importance of potential evapotranspiration in their intended context when choosing a drought index. The stark differences between SPI- and

SPEI-based projections highlight the need to understand the interplay between increasing atmospheric evaporative demand and moisture availability under a changing climate.



## 1 Introduction

Anthropogenic climate change is already affecting the frequency and intensity of droughts on all continents, through increases
in atmospheric evaporative demand (AED) and in some regions, also through precipitation (Seneviratne et al., 2021). How
much larger these changes become depends on current and future emissions. Droughts can have enormous damaging societal,
economical and ecological impacts, so understanding these events and their climate change induced changes is important.
Drought definition has been declared by Yevjevich (1967) one of the principal obstacles to investigation of droughts. Distilled
to its most simple form, a drought can be defined as a deficit of water relative to normal conditions (Sheffield et al., 2012).
Four different types of drought are typically recognised, based on where this deficit takes place (Wilhite and Glantz, 1985). A
meteorological drought indicates a period of below-normal precipitation, while a soil moisture drought (also called agricultural
drought) has below-normal levels of soil moisture availability for plants. These conditions can then cause low flows in rivers
or low water levels in lakes, called hydrological drought (of which groundwater drought can be considered a sub-type). These
types of drought can lead to ecological and socio-economic impacts, the latter often referred to as a socio-economic drought.
Whereas precipitation is the only atmospheric variable informing meteorological drought, for agricultural and hydrological
droughts AED comes into play as well, as high AED can aggravate the effects of sustained precipitation deficits through
physical and plant physiological processes (Vicente-Serrano et al., 2020b). AED is the potential of the atmosphere to evaporate
water, and is influenced by air temperature, net radiation, humidity, pressure and wind speed. It is frequently represented using
potential evapotranspiration for a reference crop (Allen et al., 1998), which leaves only the effect of atmospheric variables.
Identifying and quantifying drought conditions requires further narrowing down of the drought definition, as a universal
quantitative definition of some general state of drought would be impractical (Lloyd-Hughes, 2014). This is typically done
using one or more drought indices and one or more threshold values, where the actual condition is classified as drought when
a threshold is crossed and the value of the drought index reflects the severity of the drought. A large number of drought indica-
tors can be found in literature (Keyantash and Dracup, 2002). Drought indices that only rely on atmospheric data are a popular
choice due to data availability and propagating model uncertainties. The Drought Severity Index (DSI, Phillips and McGregor
(1998)), for example, uses precipitation only and has been used in previous studies on the impact of climate change on drought
in the UK (e.g. Blenkinsop and Fowler, 2007; Rahiz and New, 2013; Hanlon et al., 2021). The SPI (McKee et al., 1993) is a
widely used precipitation-based drought index that is recommended by the World Meteorological Organisation (Svoboda et al.,
2016). It is one of the variables shown in the UK Water Resources Portal (https://eip.ceh.ac.uk/hydrology/water-resources), and
has been used in earlier work on drought under climate change in the UK (e.g. Vidal and Wade, 2009; Arnell and Freeman,
2021). Since the introduction of the SPI, many other standardized indicators have been developed that apply the SPI standard-
ization procedure to different (combinations of) drought-relevant variables. The Standardized Precipitation Evapotranspiration
Index (Vicente-Serrano et al., 2009) gives the anomaly in a simple climatic water balance, computed as the difference between
precipitation and potential evapotranspiration (PET). Contrary to the SPI, SPEI is not an indicator of pure meteorological
drought, but instead an atmospheric-based index that reflects an upper bound for the overall water-balance deficit, most closely
reflecting the actual water balance in humid regions (Seneviratne et al., 2021). Another very widely used drought indicator that



combines precipitation and PET is the Palmer Drought Severity Index (PDSI) (Palmer, 1965). The different temporal scales of the SPI and SPEI are often used to represent drought experienced in different stages of the hydrological cycle and in water resources with different degrees of sensitivity to short- and long-term water shortages. Such uses of these indicators as proxies
for other drought types requires care (Lloyd-Hughes, 2014), but can nevertheless be useful in practice.

In this study, we aim to answer the following questions.

1. Based on atmospheric-based standardized drought indices, how are drought and extreme drought frequency, duration, extent and seasonal timing expected to change under different global warming levels?

2. What is the contribution of the changes in PET and precipitation to the changes in these drought characteristics?

3. How sensitive are the projected changes in drought characteristics to the choice of atmosphere-based drought indicator, as a source of uncertainty?

To this end we identify and characterize droughts and their projected changes in the most recent ensemble of regional climate projections for the UK, using both SPI and SPEI (hereafter, SI for standardized indicators). We compare projected drought characteristics for both indices, to identify the role of changing PET, and we isolate the contribution of increasing temperature
on the contribution of PET on drought characteristics. This helps further understand the nature of droughts in Great Britain (GB) under different levels of global warming, and assess the importance of the drought index choice for climate change impact studies and stakeholder usage.

## 2 Data

### 2.1 Observations

Datasets of PET and precipitation observations were needed for evaluation, bias correction of the UKCP18-RCM, calibration of SI and calculation of historical SI. The CHESS-PE (Robinson et al., 2020) and HadUK-Grid (Hollis et al., 2019) datasets were used for PET and precipitation respectively. Both datasets were first regridded from their native 1km resolution to the 12km resolution grid of the UKCP18-RCM, by averaging of the 1km grid cells falling in each 12km cell. A land fraction was obtained based on the proportion of 1km grid cells with observations on land within each 12km grid cell, and used to exclude
grid cells with a land fraction lower than 50% from the analysis. As no observation-based PET was available for Northern Ireland in CHESS-PE, this region was excluded from our study. The method used to obtain PET in the production of CHESS-PE is an implementation of Penman-Monteith PET for a reference grass crop (Allen et al., 1998), in which the calculation of vapour pressure deficit from temperature is based on Richards (1971) (Robinson et al., 2017).

### 2.2 The UKCP18 regional climate projections

UKCP18 is the most recent set of national climate projections for the UK. This study makes use of its third strand, a perturbed physics ensemble (PPE) of regional climate projections (UKCP18-RCM; Met Office Hadley Centre (2018)), available





from the Centre for Environmental Data Analysis. This ensemble of 12 simulations was constructed by dynamically downscaling global HadGEM3-GC3.05 simulations through one-way nesting with the same model at finer resolution. At both resolutions, HadGEM3-GC3.05 was perturbed in 47 parameters spread over model representations of convection, gravity wave drag,

boundary layer, cloud, large-scale precipitation, aerosols, and land surface interactions (Murphy et al., 2018). The horizontal resolution of the RCM simulations is 12km over GB (available on OSGB36 grid projection). Simulations of different variables are available from 1 December 1980 to 30 November 2080 on a daily time step (for practical reasons, December 1980 was left out of our analysis).

    While AED increases with rising temperatures, changes in humidity, net radiation and wind speed can also play a signifi-

cant role. Therefore, we represented AED by PET calculated using Penman-Monteith, which includes the effect of all these variables. This method leads to a more robust correlation between the resulting SPEI and soil moisture under a warming climate compared to using the temperature-only Thornthwaite method (Feng et al., 2017) and is recommended over simpler temperature-based methods (e.g. Dewes et al., 2017), however it is still subject to significant limitations (Milly and Dunne, 2016; Greve et al., 2019). The calculation of PET for the UKCP18-RCM follows the same variant of the Penman-

Monteith method used by Robinson et al. (2017), to ensure consistency with CHESS-PE. It uses these variables simulated by the UKCP18-RCM ensemble: specific humidity, pressure at sea level, net downwelling longwave radiation, net downwelling shortwave radiation, wind speed at 10m and daily average surface air temperature. PET was set to zero wherever a calculated value was negative (which occurred for less than 1% of the values overall and, when split by ensemble member and month, also less than 1% for all cases except December in ensemble member 1 with 1.2% of negative values).

To investigate the influence of the projected temperature trend on changes in SPEI-based droughts and the deviation of SPEI from SPI, we also computed an alternate version of projected SPEI ($\text{SPEI}_{dtr-tas}$) using a detrended version of UKCP18-RCM temperature. For this, a linear trend was fitted to, and subsequently subtracted from, the simulated temperature time series for each grid cell and month separately. This detrended temperature dataset was used to compute PET as described above, resulting in a $\text{PET}_{dtr-tas}$ variable in which any trend left is due to trends in other variables (specific humidity, radiation, wind

speed and pressure) or in interactions between variables. As these variables are closely intertwined in the climate models, this unavoidably introduces a physical discrepancy between temperature and the other variables used in the PET calculation. This is taken into account in the interpretation.

    As comparison to observations revealed significant bias in the simulation of both precipitation and PET, these variables were statistically post-processed using the ISIMIP3b change preserving bias adjustment method (Lange, 2019) version 2.4.1

(Lange, 2020). For precipitation, the gamma distribution and mixed additive/multiplicative per-quantile change preservation were used. For PET and $\text{PET}_{dtr-tas}$, the Weibull distribution, detrending and mixed additive/multiplicative per-quantile change preservation were used. A dry threshold of 0.1 mm day$^{-1}$ was selected below which there is considered to be no precipitation or PET.





## 3  Methods

### 3.1  Time slice selection

The UKCP18-RCM simulations used in this study are available for the RCP8.5 emissions scenario, and the models used have high global climate sensitivity compared to the CMIP5-ensemble and the probabilistic projections (Murphy et al., 2018). If fixed future time slices (e.g. 2025-2050 and 2055-2080) were to be used, the assessed changes would thus correspond to a higher level of change than what is likely to occur during that period. Therefore, to assess the impact of climate change on drought characteristics in scenarios with lower climate sensitivity and more mitigation (resulting in lower warming levels above pre-industrial times), a time slice approach was implemented. For each ensemble member, a time slice was selected from 12 years before to 12 years after the year in which the centred 25-year rolling mean global temperature exceeds 2 °C and 4 °C in the driving global model (see Table 2 in Gohar et al. (2018)). As opposed to the fixed reference period, the time periods used to represent different levels of warming are thus different for each ensemble member, depending on when their global driving models reach +2 and +4 °C. Both warming levels are reached in all 12 ensemble members, however for ensemble member 8 the time slice representing +4 °C is cut short 2 years by the end of the simulated period. This approach would result in an accurate assessment of changes in GB drought projected at these warming levels if these changes would scale directly with temperature increase (independent of the speed of change), and if the regional model has the same climate sensitivity as its driving global model. Neither of these requirements are likely to be fully met. UKCP18-RCM projects slightly weaker UK temperature responses towards the end of the simulated period than the driving global simulations (Fig. 5.2 in Murphy et al. (2018)). Also, midlatitude atmospheric circulation patterns in the selected time slices (which influence UK weather and therefore drought events) may respond to a higher level of radiative forcing than the global temperature increase levels used to select them (Ceppi et al., 2018). Nevertheless, the applied time slice approach is a reasonable approximation and frequently used for investigating impacts at different levels of global warming.

### 3.2  Drought and aridity indices

To assess how the average aridity of UK regions changes in the climate projections being used, the aridity index was calculated as the annual average ratio of precipitation to PET.

Multi-scalar standardized climate indicators allow for comparison of unusually dry (or wet) periods across locations with different climates. The Standardized Precipitation Index (SPI, McKee et al. (1993); Mckee et al. (1995)) is commonly used for characterizing meteorological drought (eg. Barker et al. (2016)), defined as a period of below-normal precipitation levels. To include the effect of other meteorological variables that play a role in drought by influencing the AED, such as temperature, the Standardized Precipitation Evapotranspiration Index (SPEI) was developed later, following the same basic principle as the SPI but using the difference between precipitation and PET Vicente-Serrano et al. (2009). SPI and SPEI are calculated following recommendations provided in Stagge et al. (2015b), using aggregation periods of 3 to 24 months. A 50-year period (1961-2010) of observation-based data (regridded HadUK-Grid and CHESS-PE) was used to calibrate the SPI and SPEI calculation. This observation-based calibration was also applied to the UKCP18-RCM data to allow a direct comparison of the results





between climate model ensemble members and observations. This is only appropriate because the bias adjustment brings the distributions of the climate model data close to the observed distributions. Classes of dry or wet period severity can be derived from SI values using thresholds introduced in McKee et al. (1993). More specifically, we consider a category of "all/total 155 drought" covering all SI of -1 and lower, and a category of "extreme" drought covering SI values of -2 and lower. Note the difference in terminology with the context of drought risk planning in the UK water industry, where "extreme" indicates return periods over 500 years.

### 3.3 Drought characterisation

Given the importance of both space (e.g. extent, spatial connectivity, local vulnerability) and time (e.g. seasonal timing, du-
ration) for drought impacts, the spatiotemporal characterisation of droughts is an important element of any drought study. It is approached here in three ways. First, the frequency of dry and extremely dry conditions was computed for each individual grid cell of GB, for each ensemble member and the observations. A distinction is made between 'extreme' droughts (SI < -2) and 'all' or 'total' droughts (SI < -1), which includes moderately, severely and extremely dry conditions. Second, the area fraction simultaneously in drought was computed to represent drought extent. The frequency distribution of this drought extent
metric is then computed for extremely dry and all dry conditions. Third, regionally averaged SI values were used to investigate drought duration and seasonality. The UKCP18 administrative regions (ukcp18 data, 2021) shown in Fig. 1 were chosen for this purpose, as they represented a decent trade-off between the number of regions to compare and relevant differences in climatology, projected changes and societal relevance. Figs. for London and the Isle of Man are not shown because of their comparatively small areas.

Distinct drought events are defined as periods of continuously negative SI values reaching a threshold value of -1 or lower. Extreme droughts are then identified as events that have a peak (i.e. minimum) SI value below -2. The duration of the drought is then defined as the length of the period during which the SI is negative. To assess changes in drought duration and the occurrence of multi-year droughts, SI computed with an aggregation period of 6 months was used. This sub-yearly aggregation period is frequently used (e.g. Stagge et al. (2017); Parsons et al. (2019)), and ensures any resulting drought durations of a year
or longer have been sustained throughout all seasons. The peak intensity of a drought event is defined as the minimum SI value reached during the drought. This was capped at -3 to limit the uncertainty induced by extrapolating into the very extreme tails of a distribution fitted to the relatively short time series available (Stagge et al., 2015b). A continuous drought event is assigned to the time slice (reference period, +2 °C or +4 °C) that contains its central time step.

## 4 Projected climatic changes

Regionally averaged seasonal cycles of precipitation (blue) and PET (yellow) are shown in Fig. 2. In all regions, existing seasonal patterns become more pronounced under a warming climate. The PET seasonal cycle becomes amplified, with relative increases noticeable from the start of spring to September, especially in the East and South. In winter, there is less agreement on relative changes of very small PET values. In summer, in the East and South, the combination of increasing PET and



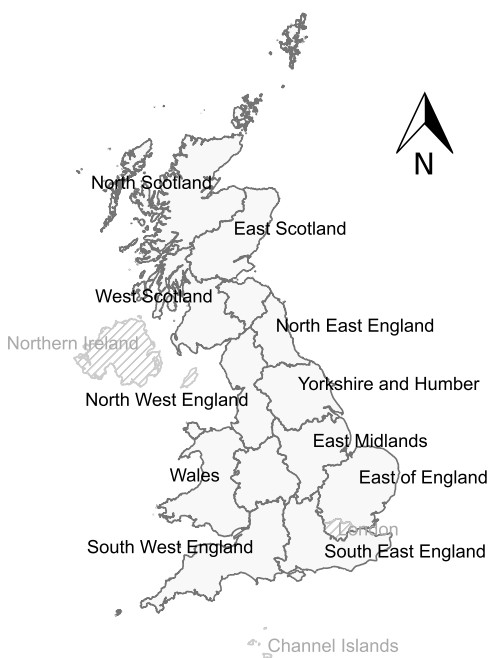

**Figure 1.** Map of administrative UKCP18 regions used for regional drought characterisation

decreasing precipitation lead to an increasing gap between the two, and an increasing period where atmospheric demand for

moisture exceeds supply (light yellow area). In some areas (e.g. North or West Scotland), the reference period precipitation

exceeds PET year-round (light blue area), but a warming climate causes there to be periods in summer where this is no longer

the case. In most months, there is a larger ensemble spread in the simulated changes of precipitation than of PET.

Large parts of GB are projected to become more arid in most ensemble members (Fig. 3). This is especially the case in the

(South-) East and Midlands, where in the reference period the aridity index was already close to 1 (annual PET roughly equal

to annual precipitation) and PET starts to exceed precipitation on an annual basis under a +2 °C warming scenario. While the

ensemble agrees very well on the spatial patterns of aridity increases, there is significant ensemble spread in the magnitude of

change. In the +4 °C scenario, widespread aridification in East, North- and Southeast England and the Midlands is predicted in

all ensemble members, but only three ensemble members simulate small areas in the East and South East crossing the threshold

of a dry sub-humid climate (aridity index < 0.65). The south coast is also projected to become more arid, but there the aridity

index does not fall below 1 even at +4 °C, except in three ensemble members and then only in the far east of the south coast.

In other parts of the UK, the aridity index undergoes no changes or shows small changes towards a more arid climatology.





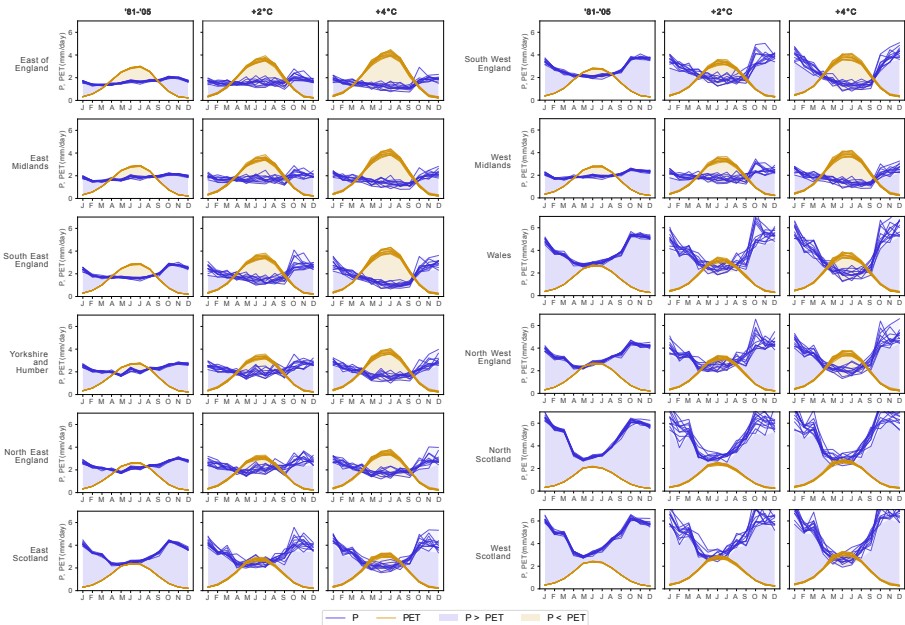

**Figure 2.** Seasonal cycle of precipitation (P; blue lines) and potential evapotranspiration (PET; orange lines) for the 12 UKCP18 ensemble members, after bias correction using change preserving quantile mapping (Lange, 2019). The different lines represent different ensemble members.

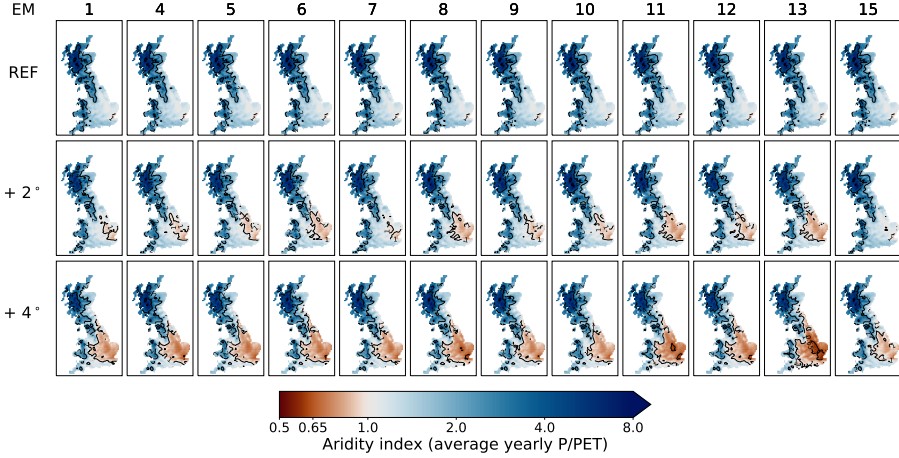

**Figure 3.** Aridity index (average annual P/PE) for the 12 UKCP18 ensemble members, after bias adjustment using change preserving quantile mapping (Lange, 2019). The contours shown in black are powers of 2 and the level of 0.65, below which a climate is classified as dry sub-humid.



## 5  Projected changes in drought characteristics

### 5.1  Drought frequency

Figure 4 shows the spatially averaged frequency of dry and extremely dry conditions based on SPI6 and SPEI6 for three time
slices representing different warming levels. Considering a GB average, the UKCP18-RCM ensemble projects an increased
frequency of moderate to extreme drought conditions in all cases under the higher warming level, and in nearly all cases under
+2 °C warming.

Using SPI6 (purely meteorological drought) with a 6-month accumulation timescale, at global warming of 2 °C above
pre-industrial, on average there is a moderate increase in total drought frequency, with a larger relative increase in extremely
dry conditions. There is considerable ensemble spread on the magnitude of these changes, with three (one) ensemble members
projecting a decrease in total drought and extreme drought frequency. At 4 °C above pre-industrial levels, all ensemble members
agree on increased total meteorological drought frequency, ranging from a few percent points (ensemble member 4) to more
than double the original frequency (ensemble members 6, 12 and 13). For extreme meteorological drought, all ensemble
members project multiples of the reference period frequency by +4 °C. Most ensemble members project greater changes in
meteorological drought between +2 °C and +4 °C than between the reference period and +2 °C, which is not unexpected as
the reference period is already warmer than pre-industrial levels. Over half of the ensemble members (1, 7, 8, 9, 10 11, 15)
project a very small change of a few percent in total meteorological drought frequency by +2 °C, followed by a large increase
between +2 °C and +4 °C. In contrast, there are two ensemble members (6 and 12) that project the largest increase in drought
conditions between the reference period and +2 °C of warming, followed by a small additional increase by the +4 °C time
slice.

Using SPEI6 (and thus including PET increases as well as precipitation changes), all ensemble members agree on monoton-
ically increasing extreme and total drought frequency with progressive global warming. By +2 °C, the ensemble average gives
a doubling of spatially averaged SPEI6-based drought frequency and an increase of extremely dry conditions from 2% to 9%
of the time. Under +4 °C of global warming, the ensemble projects SPEI6-based drought almost half of the time (ensemble
average: 46%), about half of which (ensemble average: 23%) are classed as extremely dry conditions in today's climate (SPI
and SPEI were calibrated with observations from 1961 to 2010). The total drought frequency increase using SPEI still tends to
be greater going +2 °C to +4 °C than going from the reference period to +2 °C, though the contrast is less than for SPI6.

The GB-averaged drought frequency increases are, for each ensemble member and warming level, greater using SPEI than
using SPI. This is the case for all severities combined as well as for extremely dry conditions. For ensemble members that
project a slight decrease in drought frequency using SPI, including PET in the drought indicator (SPEI) changes the sign of the
projected change. For 10 ensemble members considering all drought, and for 6 when considering extreme drought, the SPEI6-
based GB-average drought frequency projected at +2 °C is equal to or greater than the SPI6-based frequency projected at +4 °C.
The scatter plots in Fig. 4 show the relationship between GB-averaged drought frequencies using SPI6 and SPEI6 as projected
at different GMWL in the 12 ensemble members. Projected drought frequencies all lie above the identity line, confirming that
increases in SPEI6 drought frequency exceed increases in drought frequency defined by SPI6. All points move upwards (more



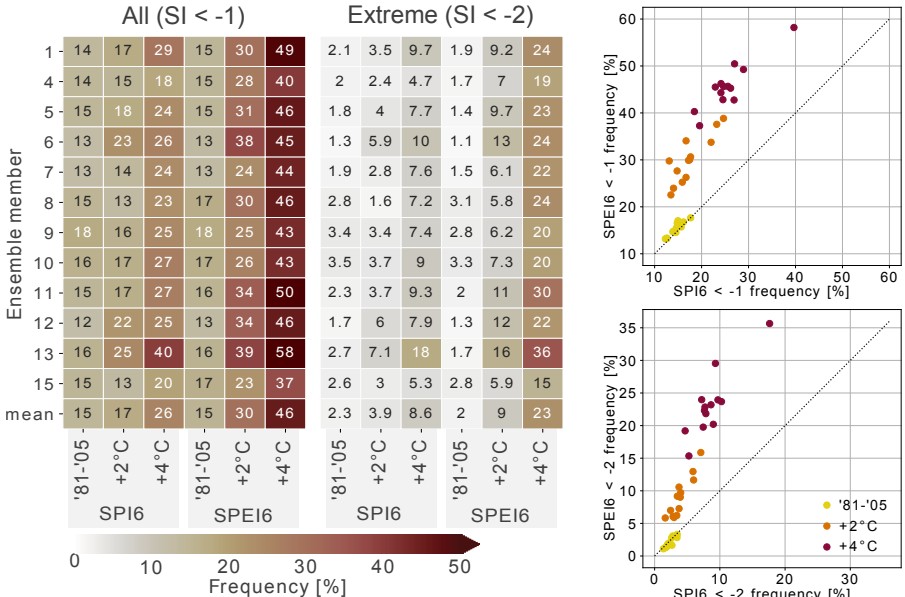

**Figure 4.** Spatially averaged projections of drought frequency for each ensemble member (rows) and the ensemble mean (bottom row), for three time slices (subcolumns) that correspond to different levels of global mean surface warming compared to pre-industrial levels, using both drought indices (columns). Frequencies shown for all droughts (left), extreme droughts (middle), and as scatter plots (one point per ensemble member) comparing SPI and SPEI frequencies for all droughts (top-right) and for extreme droughts (bottom-right). Spatial averages are across the whole of Great Britain. Heatmap colourmap: bilbao (Crameri, 2018)

frequent SPEI6 events) with increasing global warming level, and most move to the right (more frequent SPI6 events) except for a few ensemble members for +2 °C noted above. The ensemble spread (scatter) grows with increasing warming level and there is an indication of a nonlinear relationship (due to SPEI6 occurrences beginning to saturate when they have already become quite frequent). Finally, it is interesting to note that the SPI6 and SPEI6 frequencies are not perfectly correlated, indicating

that changes in precipitation and PET are not necessarily highly anticorrelated amongst ensemble members (i.e. an ensemble member simulating a strong decrease in precipitation may not simulate a greater increase in PET).

  The relative increase of extreme drought frequency is much greater than the relative increase of the total drought frequency. By +4 °C, the ensemble mean spatially averaged total drought frequency increases by a factor 1.7 for SPI and by a factor 3.1 for SPEI. For extreme droughts, however, these multiplication factors are 3.7 and 11.5, respectively. This larger relative

increase for more extreme events is important, and it has been noted before that even small changes in the mean can be paired with amplified changes in low-probability events (e.g. Vicente-Serrano et al., 2020a).

  The maps in Fig. 5 show the spatial patterns of these drought frequency changes (for the ensemble average) and their differences between SPI6 and SPEI6. For the reference period, the ensemble-averaged GB mean total and extreme drought frequencies are 0.15 and 0.023 respectively, which are close to the theoretically expected values of 0.158 and 0.022. There is some

variation around these values in space (Fig. 5) and among ensemble members (Fig. 4). This is expected. First, imperfections in





the fit of the gamma and GEV distributions used in the calculation of SPI and SPEI in the very dry tail of the distribution will result in deviations from the theoretically expected frequencies. Second, even if the distributions fit perfectly, they were not fit to the reference period of the simulations, but rather to 50 years of observed data. Differences between the climates of the 1961-2010 and 1981-2005 periods, model errors remaining after bias adjustment and internal climate variability can all result

in differences between the simulated drought frequency in the reference period and the theoretical frequencies that would be expected for the calibration data.

There is significant regional variability in projected drought frequency across GB using either drought indicator, especially for extreme drought (Fig. 5). At +2 °C, SPI6 projects a relatively homogeneous increase in drought frequency over GB, with slightly greater increases near high elevations. Greater regional contrasts emerge for SPI6 at +4 °C, with respectively lesser

and greater drought frequency increases on the west and east of highly elevated areas, and decreased drought frequency along the northwest of Scotland. The regional patterns of projected drought frequency for SPEI6 also show features apparently linked to topography or proximity to the west coast, where drought frequencies show smaller increases. Regional contrasts are the largest for SPEI under +4 °C above pre-industrial levels, with limited change of either sign in drought frequency in northwest Scotland but strong increases across most of the rest of GB, including the East Midlands and East England where drought

conditions are projected around 60% of the time. While the projections of SPI and SPEI follow similar broad spatial patterns, the areas projected to experience the greatest increase in frequency of dry conditions differ between the drought indices. For SPI, increasing all-drought and extreme-drought frequencies are most apparent on the eastern sides of the west coast upland areas. For SPEI, these areas as well as a larger area stretching from the West Midlands to the East of England experience the greatest increases in drought frequency, which shows that the difference between drought indices is region dependent. For both

indices, isolating extreme droughts amplifies these regional patterns of change.

The bottom row of Fig. 5 shows SPEI6$_{dtr-tas}$, which is the SPEI6 using PET calculated with detrended temperature simulations. This row is compared to the ones above to estimate the contribution of temperature to the SPEI6-based drought changes shown. Without the projected temperature increase, SPEI shows only minor changes in all or extreme drought frequency. The changes are much smaller than those with the standard SPEI6, especially for extreme droughts, indicating the dominant

role of warming in driving more frequent drought. Furthermore, at +4 °C, the projected drought frequencies with detrended temperatures are much less than those found for the precipitation-only SPI6. On the face of it, that suggests non-temperature influences may reduce PET (offsetting some of the temperature-driven increase) and that purely temperature-based PET might overestimate drought risk based on these future projections. Zhao et al. (2021) also found that other variables partially offset the effect of warming temperatures on the PET component of SPEI droughts, although in a very different climatic setting.

However, the effects of physically inter-dependent variables (especially temperature and humidity) cannot be truly separated. Crucially, here we use simulated specific, not relative, humidity to compute PET (Robinson et al., 2017). Whereas specific humidity is projected to increase over GB, relative humidity is projected to decrease as the saturated humidity increases faster with rising temperatures (not shown), contributing to the increased future PET. Detrending the temperature but leaving the projected specific humidity increase unchanged, leads to increasing relative humidity and decreasing vapour pressure deficit

computed in the aerodynamic component of Penman-Monteith PET, and a decrease in PET. Therefore, the difference between





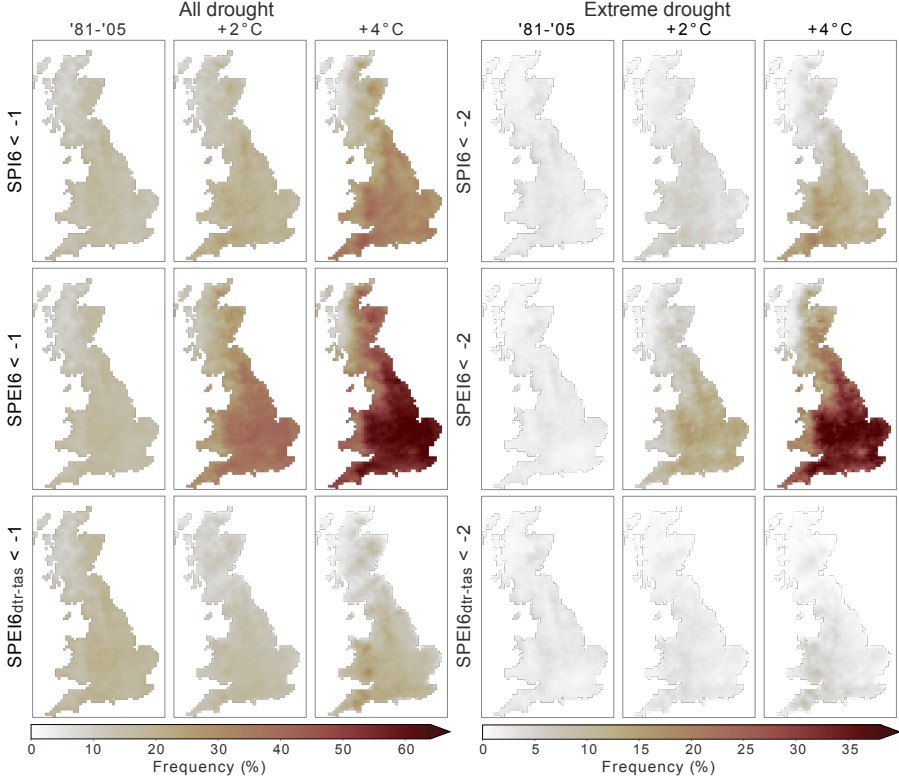

**Figure 5.** Ensemble averaged projected frequency of all (left) and extreme (right) dry conditions. Top: SPI6, middle: SPEI6 with projected temperature changes, bottom: SPEI6 with detrended temperature. Colourmap: bilbao (Crameri, 2018)

SPEI6 and SPEI6$_{dtr-tas}$ combines the direct effect of temperature detrending and an indirect effect from creating an artificial downward trend in vapour pressure deficit. The temperature effect shown by the SPEI6 - SPEI6$_{dtr-tas}$ difference (Fig. 5) thus encompasses more than if a PET formulation using relative humidity would have been used. Robinson et al. (2017) quantified the contributions of different atmospheric variables to changes in AED over GB from 1961 to 2012 using specific

and relative humidity-based PET formulations and found that using relative humidity significantly decreases the contribution of air temperature compared to using specific humidity. When using specific humidity, they found that temperature is the dominant variable leading to increasing AED (largely through its influence on the aerodynamic component) but its overall effect and its increasing influence on the aerodynamic component are smaller than that of relative humidity when using a relative humidity-based PET formulation. Based on CMIP5 projections, Fu and Feng (2014) found that decreasing relative humidity has a major

influence (37%) on changes in global aridity over land, albeit smaller than the contribution of temperature (53%).





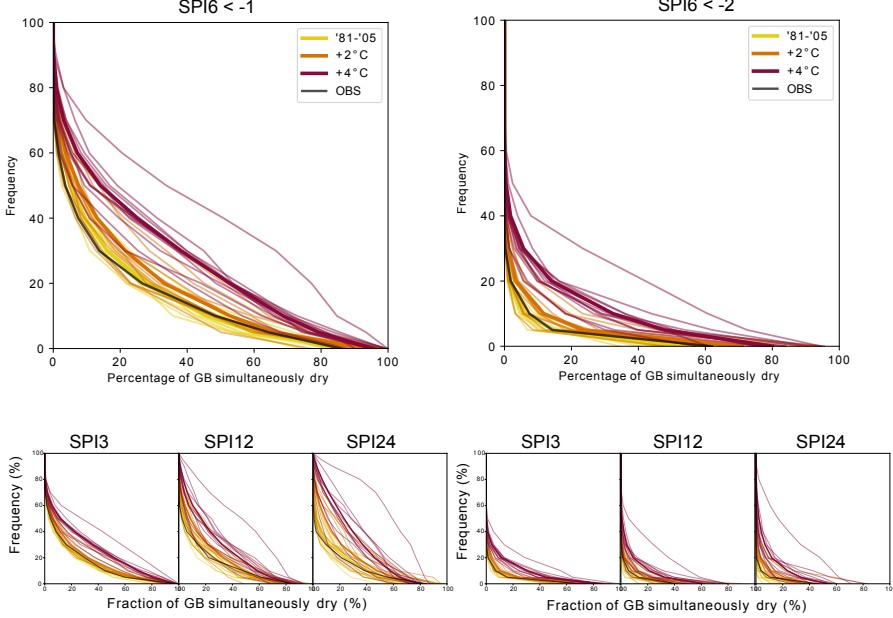

**Figure 6.** Extent-frequency curves for all (left) and extreme (right) drought extents based on SPI at different aggregation levels (subplots). The horizontal axis gives the drought extent (as fraction of GB area) that is reached or exceeded with a frequency given by the vertical axis.

## 5.2 Spatial extent

Figure 6 and Fig. 7 show the observed and simulated extent-frequency curves of drought conditions for SPI and SPEI respectively, for different global warming levels (i.e. time slices) and using different aggregation levels. Moving upwards in this plot means an increase in the frequency of drought conditions with at least the spatial extent given by the horizontal axis (not

necessarily in the same locations). Moving to the right in this plot means an increase in the spatial extent of drought conditions that is exceeded with a particular frequency (given by the vertical axis).

   The relationship between frequency and drought extent for the reference period simulations generally match well between reference period simulations and observations for both SI, for all aggregation periods shown. However, the match is not perfect. As the aggregation period increases, the frequencies of smaller drought extents are increasingly overestimated in the simula-

tions while the frequencies of larger drought extents are on average well represented (SPI) or become slightly underestimated (SPEI). The frequencies of larger drought extents are well-represented for the sub-yearly aggregation periods, but tend to be underestimated using SI12 and SI24, with a larger underestimation for SPEI than for SPI. This could indicate an underestimation of spatial coherence of the simulations, which was not considered explicitly in the bias adjustment.

   For a given fraction of the GB simultaneously dry, the relative change in frequency as global temperature increases is far

greater for extreme droughts than for all droughts (for both SPI and SPEI). For instance, based on SPI6, the frequency that at least 20% of GB simultaneously experiences a drought is 26% currently, and 44% with +4 °C of warming (mean of the

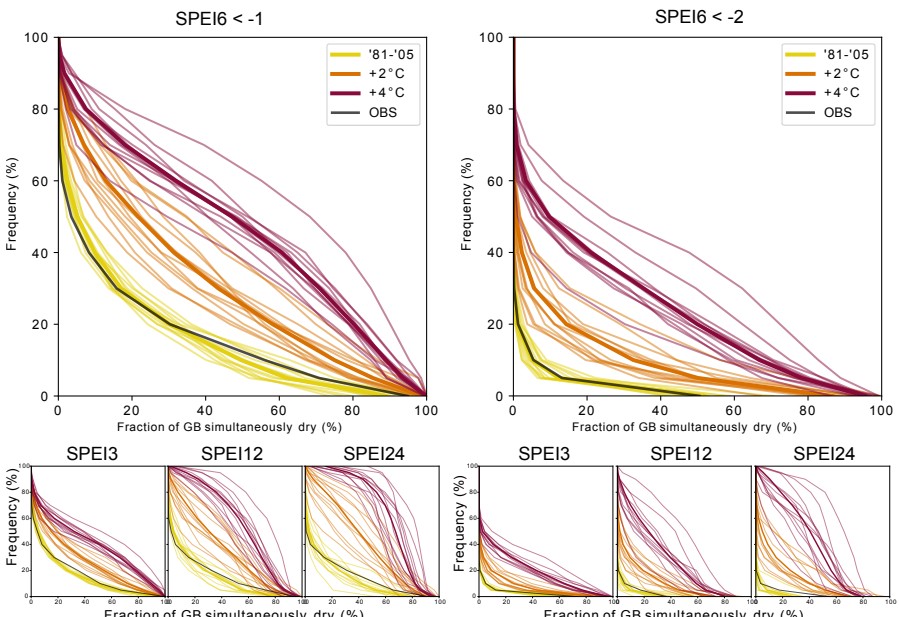

**Figure 7.** As Fig. 6 but for SPEI rather than SPI

ensemble). In contrast, extreme droughts covering 20% GB have a frequency of occurrence currently less than 4% of the time, but this frequency is expected to jump to 16% for a +4 °C warming (mean of the ensemble). Using longer aggregation periods, the ensemble spread increases, the difference between GMWL become more obvious and differences between drought

indicators are more pronounced.

The fraction of the time that nowhere in GB is experiencing dry or extremely dry conditions, given by the difference between 100% and the intercept on the y-axis on Fig. 6 and Fig. 7, matches closely between the reference period observations and ensemble averaged simulations. The drought-free frequency is generally projected to decrease under climate change. This change is far more drastic based on the SPEI than SPI. Using ensemble averages for the 6 month aggregation period, the SPI

gives a reduction in drought-free frequency from 17% in the reference to 14% by +2 °C and 12% by +4 °C, and a reduction in extreme drought-free frequency from 58% to 54% by +2 °C and 44% by +4 °C. With the SPEI, this becomes a reduction from 18% in the reference period to 7% by +2 °C and 4% by +4 °C for drought conditions of all severities, and from 63% to 40% by +2 °C and 23% by +4 °C for extreme drought. Using SPEI with longer aggregation periods there would always be drought somewhere by +4 °C. These large reductions using the SPEI might in part be explained by the more spatially homogeneous

increase in PET compared to precipitation, combined with the overall greater drought frequency based on SPEI.

The maximum drought extent (intercept x-axis on Fig. 6 and Fig. 7) is in most cases greater for shorter aggregation periods, although not consistently in the observations: the maximum observed extreme SPEI drought extent is greater using 24 months than using 12 months as aggregation period. The ensemble-averaged maximum fraction of the GB area in at least moderate drought is projected to increase only slightly under global warming based on SPI and SPEI. For extremely dry conditions,





however, the maximum extent is projected to increase greatly with global warming. The ensemble mean maximum SPI6 area fraction in drought increases from just over 51.2% (an underestimation of observation-based maximum extent) to just over 71.1% by +2 °C and to 80.0% by +4 °C. For SPEI6, the ensemble-averaged simulated maximum extent and the overall frequency-extent relationship matches observations very closely, and the maximum extent is projected to increase from just over 51.8% to 86.5% by +2 °C, and to 95.4% (i.e. almost all of GB simultaneously in extreme drought) at +4 °C. The relative
increase of maximum extreme drought extent projected due to global warming is greater for longer aggregation periods, for both indicators.

Climate change-induced changes in the relationship between frequency and extent of droughts depend strongly on the drought metric used. SPI and SPEI both show increasing frequency of droughts of most extents, however the increase is much greater for SPEI. In the case of SPI (except SPI24), the ensemble average frequency of droughts with an extent over 60%
of GB changes very little under +2 °C but increases strongly under +4 °C. In contrast, the frequency of SPEI-derived droughts and extreme droughts with these extents increase strongly for each GMWL, with larger increases for +4 °C.

## 5.3   Seasonal timing

Figure 8 and Fig. 9 show the contributions that summer and winter deficits make to annual droughts according to SPI and SPEI for three global warming levels for different GB regions. The horizontal and vertical axis show SI6 for March and
September respectively, indicating how dry or wet the hydrological winter and summer were in a given year. The September SI12, indicating the dryness of the corresponding hydrological year, is represented by the colours of the dots. For example, a grey dot with coordinates (1.1, -2.2) represents a normal annual value consisting of a wet winter and an extremely dry summer. The larger symbols shown are the centroids for 5 bins of SI12 values, representing the classes "extremely dry", "dry but not extremely dry", "normal", "wet but not extremely wet" and "extremely wet".

These figures show the overall progression of drought characteristics as global warming increases: the shift towards fewer wet years (fewer blue/green symbols) and more droughts (orange/brown) apparent in all regions for SPEI and all regions except North and West Scotland for SPI arises mostly from a movement of the cloud of simulated years downwards (drier summers) but offset in some cases by a movement to the right (wetter winters). In all regions, the increase in the proportion of dry years (SI12) is greater based on SPEI than based on SPI, due especially to the stronger transition to very dry SPEI summers caused
by increased PET.

In every region (except possibly North Scotland), there is a (mostly large) increase in dry summers with increasing GMWL using the SPI. This shown by the point clouds moving down, yielding an increased fraction of points below -1. This increase in dry summers is consistently much greater using SPEI, in all regions. Using the SPEI about half the summers are dry by +2 °C and the vast majority of summers are dry by +4 °C (with almost half or over half of the summers extremely dry, i.e. below -2 on
the vertical axis) in almost all English regions (East of England, East and West Midlands, South East and South West England, North West England, Yorkshire and Humber) and Wales, and to a lesser degree in North East England and East Scotland. These changes are less strong using the SPI, with changes by +4 °C above pre-industrial levels in many regions comparable to the changes by +2 °C using SPEI. In most regions, there is little or no change in the proportion of dry winters (with values below



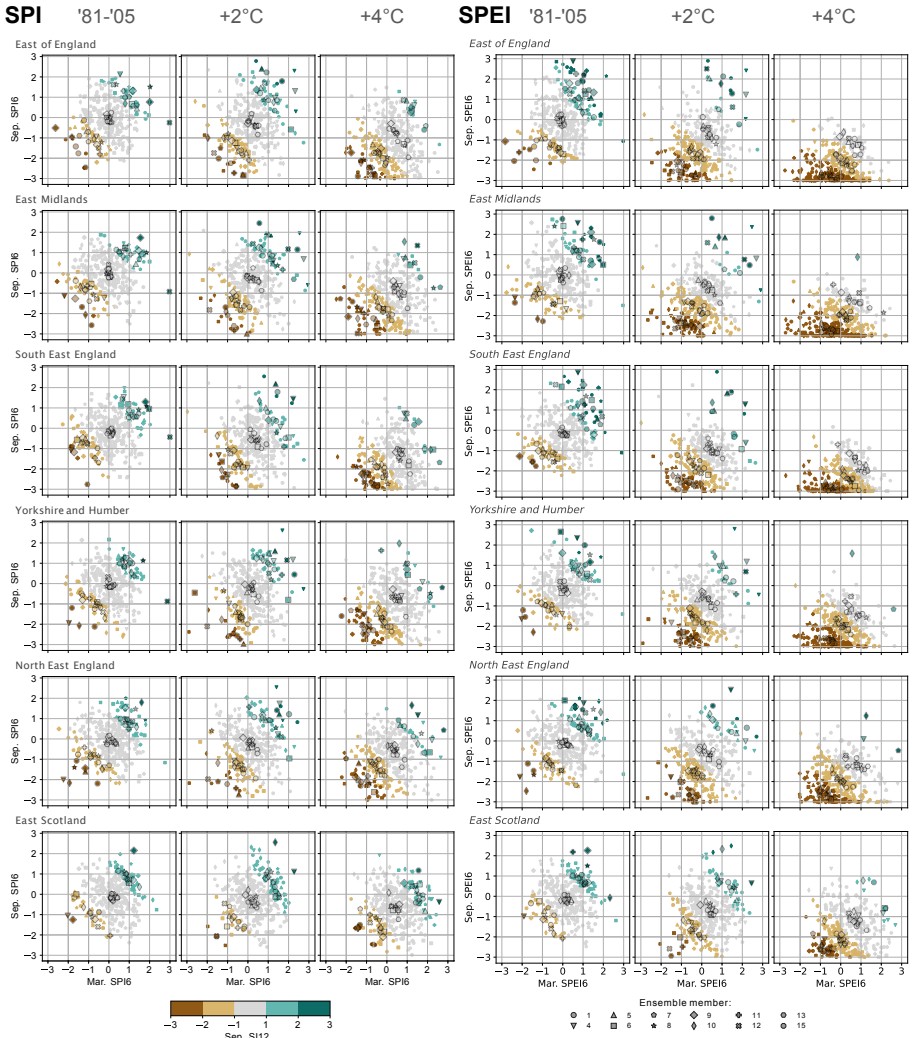

**Figure 8.** Values of September SI12 (hydrological year) plotted against the September SI6 (hydrological summer) and March SI6 (hydrological winter) from the same year used to compute SI12 for SPI (left) and SPEI (right). All years are shown for each time slice and ensemble member. SI6 values that exceed -3 or +3 are plotted at -3 or +3. The larger, transparent markers show the centroids of 5 SI12 classes: extremely dry, dry but not extremely dry, normal, wet but not etremely wet, extremely wet. Continued in Fig. 9 for other regions. Colours based on Brewer et al. (2013)

.

-1 on the horizontal axis) with either index, but when they do occur they are seen to more often lead to annual droughts because

they are more often followed by a dry summer. In West Scotland and North West England, there are fewer dry winters using either index.



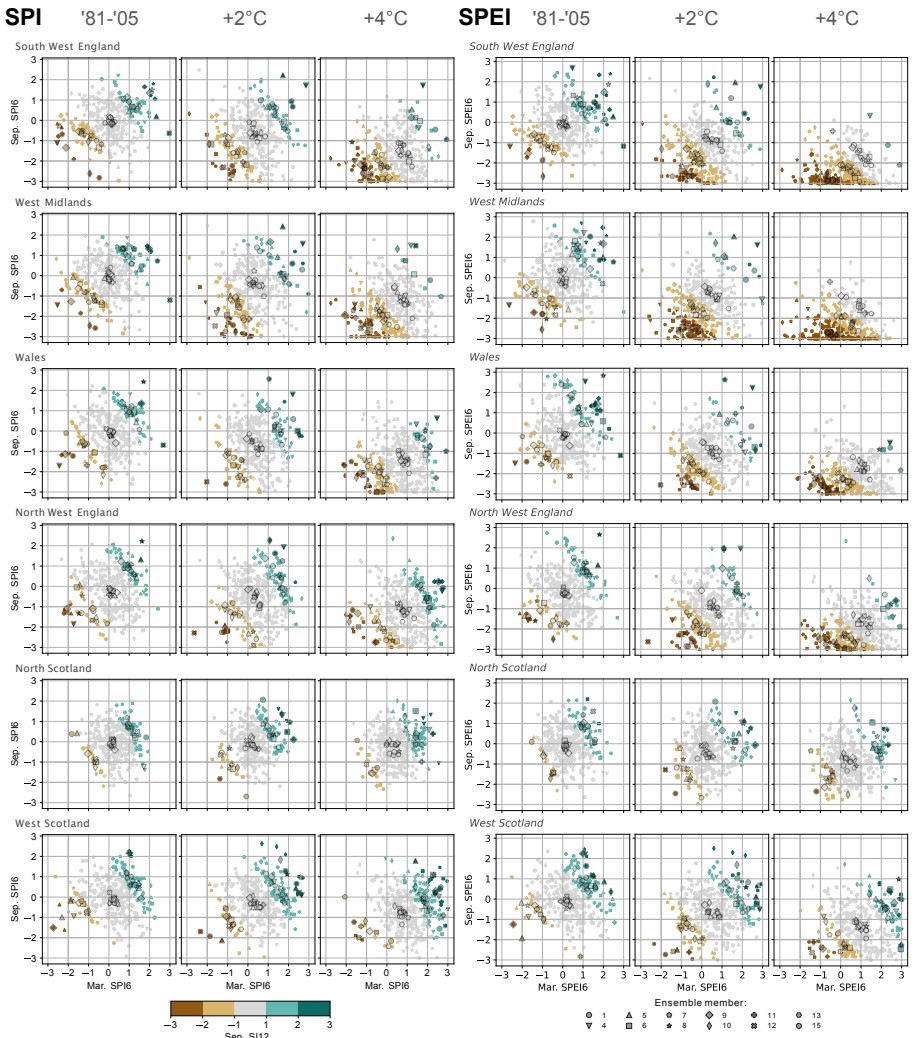

**Figure 9.** Continuation of Fig. 8.

The increasing proportion of dry years in the warming climate tends to consist of more contrasting winter and summer seasons, moving towards more intense summer droughts (down) preceded by normal or even wetter than average winters in several regions (middle/right). In the reference period, years with dry summers and wet winters (bottom right corner where SI-March > 1 and SI-September < -1; not that dry and wet are relative to "normal" conditions in standardized indices such as these) are quite rare, and almost always result in years within one standard deviation of average moisture conditions. The number of these years simulated by the ensemble increases with GMWL in almost all regions for both SI, and remain predominantly within 1 standard deviation of average moisture conditions for the SI12. However, with increasing GMWL, in all English regions plus Wales there is also a growing proportion of these wet winter/dry summer years that end up as overall dry (sometimes extremely






dry) hydrological years using SPEI (and exceptionally SPI in some regions). In some regions, a growing proportion of these years end up as overall wet with increasing GMWL, especially if the winters are extremely wet (SI6>2). This is the case using SPI in Wales (and South West England), using the SPI and to a lesser degree SPEI in North West England, using the SPEI and to a lesser degree SPI in North Scotland, and using both SI in West Scotland.

### 5.4   Duration

The duration of droughts is important for drought risk management. Figure 10 and Fig. 11 show the number of simulated drought events within 6 drought duration categories (horizontal axis) that occur in three 25-year periods representing three global warming levels, based on SPI6 and SPEI6 respectively. Figure S1 and Fig. S2 (supplementary material) show these plots for only the droughts that reach extreme levels (extreme droughts).

   Overall, especially for the 1 to 23 month durations, the ensemble spread of the number of events is large, for both indicators,
and there is often a strong overlap between GMWL which is diminished when isolating droughts that reach extreme levels. Nevertheless, some trends can be noticed across the regions.

   The sign of changes in the occurrence of droughts shorter than 6 months in duration is highly dependent on the indicator and region. Based on SPI6, the number of droughts shorter than 6 months in duration in a 25-year period of simulations increases with increased GMWL in the ensemble mean and median in most regions. The sign of change is unclear in North Scotland,
North East England and East of England. In some regions, this consists of little change by +2 °C followed by an increase by +4 °C. These increases based on SPI6 are contrasted by a decrease when using SPEI6 in 6 of the 12 discussed regions (East of England, East Midlands, South East England, North East England, South West England and the West Midlands). In these regions, this decrease in < 6 month SPEI6-droughts is accompanied by greater increases in 6 to 11-month long droughts and/or multi-year droughts. In Yorkshire and Humber and South West England, the projected changes in SPEI6-based <6 month
droughts are very small, while in Wales, North West England and the three Scottish regions, the number of SPEI6-droughts shorter than 6 months is projected to increase. The regions where sub-6 month droughts are projected to increase in occurrence with increasing climate change are the regions where, on average, precipitation exceeds PET for all months of the year in the reference climate (Fig. 2). The interruption of this seasonal pattern with a few summer months where PET exceeds precipitation in an average year in a warmed climate might explain part of the increase in short droughts in these regions. The number of
short droughts that reach extreme levels is projected increase as well (Fig. S1 and Fig. S2), and for these events there is more agreement between the SI. In the eastern English regions, their occurrence is projected to stay stable between the reference period and +2 °C followed by an increase between +2 °C and +4 °C (although for SPEI6 this is less clear), while there is a monotonous increase with GMWL in the western regions using both indicators.

   Droughts with a duration of 6 to 11 months based on SPI6 are also projected to increase in occurrence under a changing
climate for the majority of the regions, except North and West Scotland. In the East of England, East Midlands, and East Scotland, this increase is only projected for global warming exceeding +2 °C, and changes are unclear for North West England and Yorkshire and Humber. However, using SPEI6 these events increase monotonously in occurrence with rising GMWL in all regions. Using either indicator, there is a more distinct increase in 6 to 11 month droughts that reach extreme levels in all



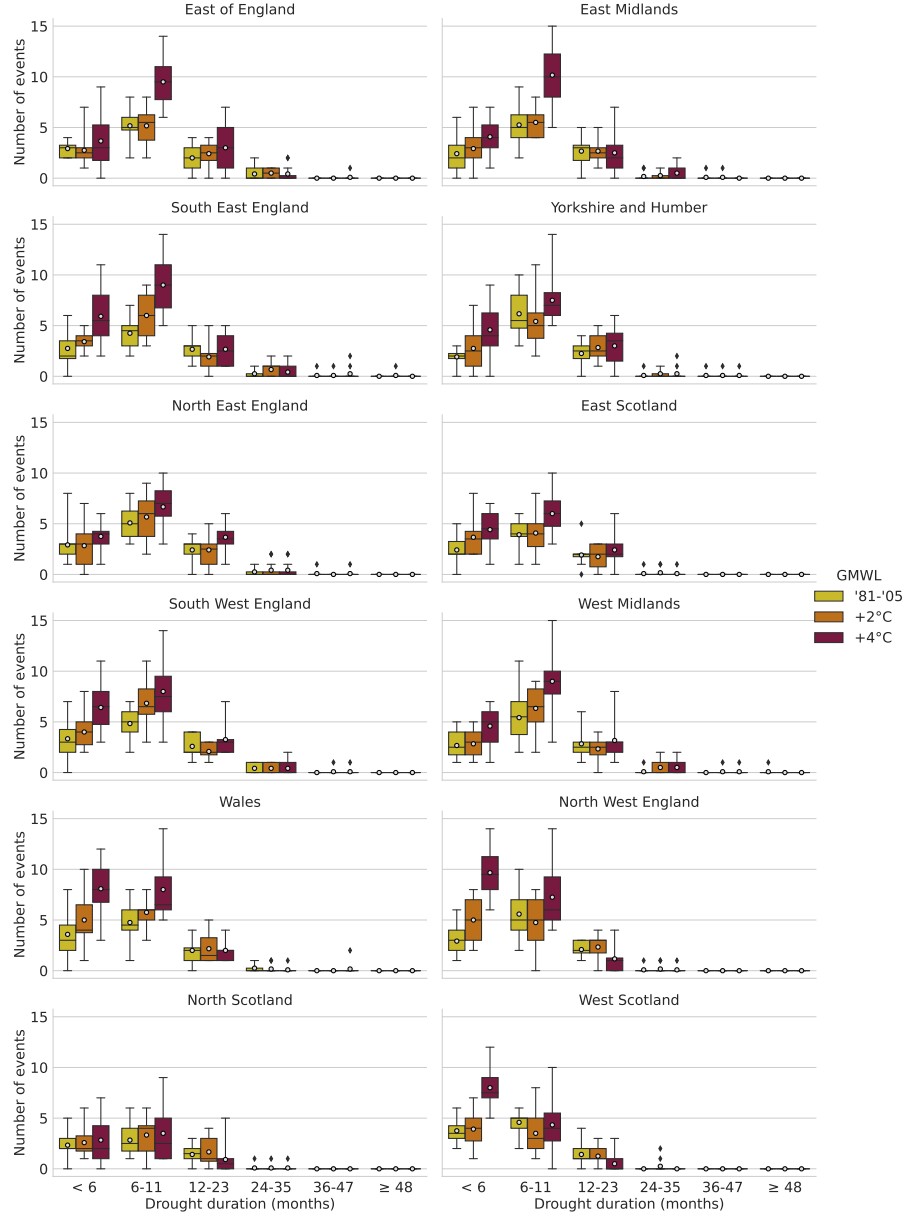

**Figure 10.** Number of droughts of all severities by duration for three 25-year periods corresponding to progressive warming scenarios in different GB regions, based on SPI6. White circles indicate the ensemble mean, boxes show the interquartile range, whiskers show the ensemble range except for members exceeding 5 x the interquartile range (diamonds).

regions (except North and West Scotland for SPI6). Therefore, a greater share of the droughts within this range of durations is

projected to reach extreme levels under climate change.

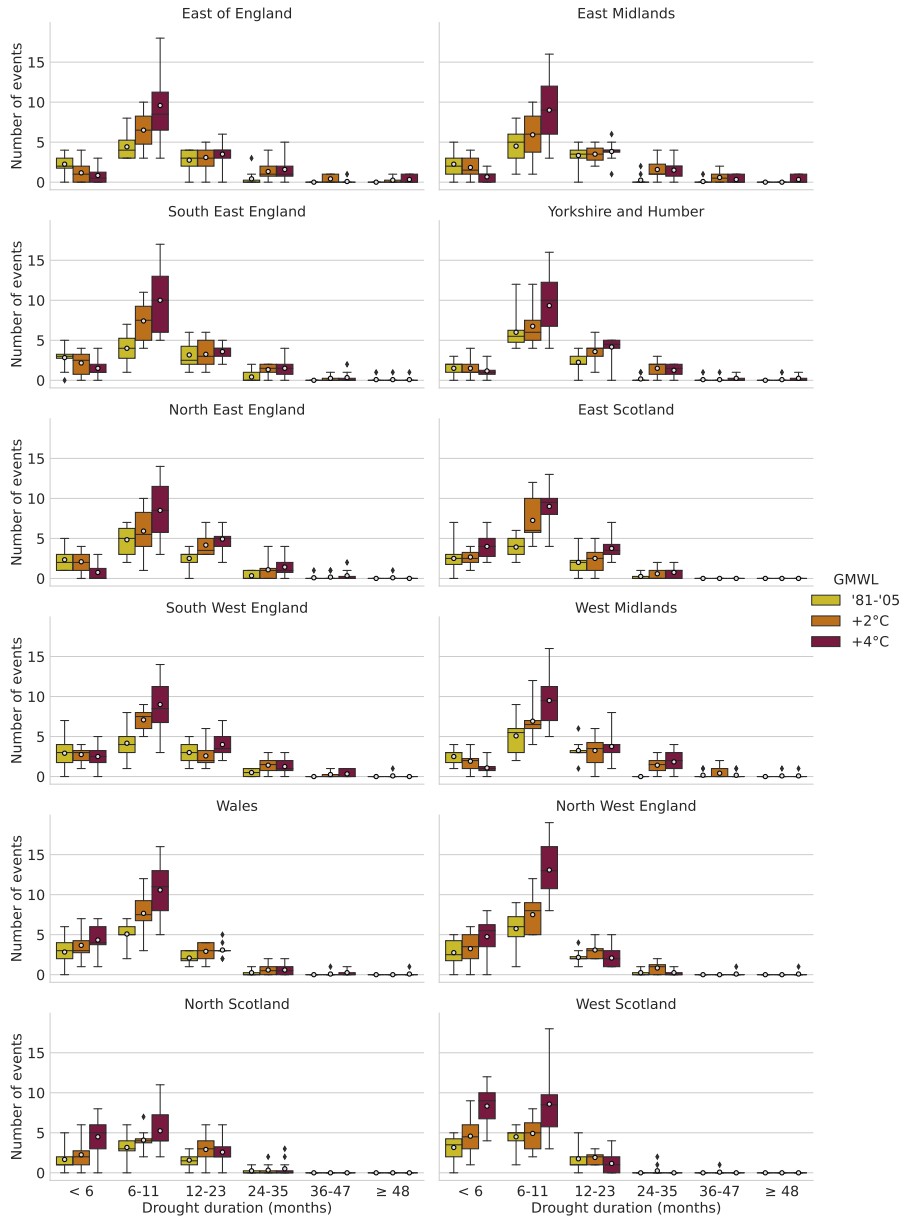

**Figure 11.** As Fig. 10, but for SPEI6.

For droughts between 1 and 2 years in duration, based on SPI6 there is little change in occurrence in many regions, with uncertain increases in some regions and decreases by +4 °C in North West England, North Scotland and West Scotland. In those regions, the number of ensemble members simulating at least one of these droughts decreases from 10-12 to 4-7. Using the SPEI6, monotonous increases in events lasting between 1 and 2 years can be seen in in Yorkshire and Humber, North East England and East Scotland, of which the former two also show uncertain increases in SPI6 drought events. Comparing this to






the results for extreme droughts indicates an increasing number of droughts reaching extreme levels for this duration category, even where the total number of events changes little (except in North West England (SPI6) and North and West Scotland (both)).

Multi-year droughts are more rare and by definition occur less in a fixed-length time slice. A greater share of the ensemble
members contains zero events for a given time slice, which makes the box plots harder to interpret, and droughts lasting at least 2 years rarely occur more than once in a given time slice in our analysis, and never more than twice for a given duration bin. Therefore, for these events Fig. 10 and Fig. 11 will be discussed jointly with the number of ensemble members that project at least 1 such event in a given time slice. Based on the SPI6, the number of ensemble members projecting at least one drought lasting from 2 to 3 years isn't projected to increase or decrease for most regions (except for possible increases in the Midlands
and decrease in Wales), although an increasing share of events reaching extreme levels is found in about half of the regions. for SPEI6. Using the SPEI6, the number of ensemble members projecting at least one event increases with GMWL in all eastern regions, South West England and the West Midlands, and for events reaching extreme levels this increases in almost all regions. The number of events simulated in a single time slice by a single ensemble member also increases in several regions using the SPEI6. Droughts lasting three years or longer in the reference period are simulated in none or one of the ensemble members
depending on the region, irrespective of the SI (with exception of the West Midlands for SPEI6: 2 ensemble members). A drought of four years or longer in the reference period is only simulated by one ensemble member in one region for each indicator. Using the SPI6, little change can be found in the number of ensemble members simulating +3 year droughts, with maximum 2 ensemble members simulating such an event in any time slice and any region. In South East England and Wales, one ensemble member simulates two such events based on SPI6, both in the +4 °C scenario and reaching extreme levels.
Based on SPEI6, however, under +2 °C and +4 °C more ensemble members simulated at least one +3 year drought event in the English regions and Wales, most of which reaching extreme levels at some point. In the East of England, East Midlands and Yorkshire and Humber, 3 or 4 ensemble members simulate a +4 year SPEI6 drought event under +4 °C, and in the East of England 3 ensemble members simulate a +4 year SPEI6 drought under +2 °C (all of which reach extreme levels). Despite the limitations of 25 year time series to investigate changes in droughts longer than 3 or 4 years, this may indicate an increased
likelihood of these multi-year droughts under a changing climate. However, as these changes are almost exclusively seen using SPEI, it depends on the importance of temperature and AED.

## 6 Discussion

### 6.1 Projected changes in atmospheric droughts

This section discusses the results presented above in the context of previous studies that have used meteorological and atmo-
spheric based drought indices to investigate climate change impacts on droughts in the GB and other regions.

Hanlon et al. (2021) used the DSI to look at changes in drought intensity and frequency in the UKCP18-RCM projections. They found increasing median drought intensity mainly in the East and South of the UK, which broadly matches the pattern of frequency increases in this study. This is also in broad agreement with the spatial pattern of increases in drought event


occurrence found by Spinoni et al. (2018) using EURO-CORDEX and a combined atmosphere-based drought indicator. While
this study identified the South West, the east of Wales and the Pennines region as expecting the largest SPI6 drought frequency
increase with global warming, the regions with the largest drought intensity increases in the 6-month aggregated DSI-based
results by Hanlon et al. (2021) are all eastern regions of Scotland and England plus the English Midlands. Bias adjustment
is unlikely to be a major cause of differences with Hanlon et al. (2021), since they used a similar bias adjustment technique
as was used in this study. Rather, these differences may be caused by differences between the drought characteristics studied
(frequency and intensity), and between the SPI and DSI. The DSI drought intensity counts any anomaly below the climato-
logical monthly average, while the SPI6-based drought frequency represents dryness exceeding one standard deviation. While
DSI values are standardized using the climatological mean rainfall, the conversion to the inverse standard normal distribution
means higher-order moments are also standardized in the SPI.

We found that the UKCP18-RCM ensemble projects an increasing frequency of droughts with smaller extents as well as
increasingly widespread droughts with increasing global warming, especially using SPEI and dependent on the aggregation
period used. This study only looked at drought extent as a fraction of GB, and not at the locations of droughts with different
extents. Nevertheless, the difference in the shape of the observation-based extent-frequency curves between extreme and all
drought conditions (Fig. 6 and Fig. 7) agrees with Tanguy et al. (2021) that the most extreme droughts tend to be less spatially
coherent, so more localised, than when all droughts are considered. A projected increase in the extent of drought and extreme
drought was also found by Rahiz and New (2013) using UKCP09 and the DSI6. As discussed in Section 6.4, widespread dry
and extremely dry conditions identified using a SI with a set aggregation period, would likely lead to different agricultural and
water resources impacts depending on the affected regions.

Drought duration is an important characteristic with respect to drought management. Previous studies have often assessed
changes in drought duration through the mean and/or median duration or overall trends (e.g. Touma et al., 2015; Naumann
et al., 2018; García-Valdecasas Ojeda et al., 2021; Vicente-Serrano et al., 2021). Here, we took a different approach by looking
at the changes in events in different duration categories, which revealed an increasing occurrence of multiyear droughts in some
regions, especially based on the SPEI. Multi-year droughts were also assessed by Lehner et al. (2017), who found that for some
studied regions (including Central Europe and the Mediterranean), progressive climate change is projected to increase the risk
of 4 consecutive drought years. They assessed this using the PDSI, which, like the SPEI is sensitive to projected increases in
PET. Rahiz and New (2013) considered changes in drought events lasting at least 3, 6, 10 and 12 months based on the DSI6.
They found widespread increases in the number of events of at least 3 months in England and part of the Scottish Highlands,
with the largest increase and ensemble agreement in Wales and South West England. This is broadly in agreement with our
SPI6-based results, however there are possible differences in the relative magnitude of the increases within England and Wales.
Moreover, the magnitude of the projected changes is generally greater in our study, particularly for drought longer than 6
months.

Seasonal timing and contributions of drought was assessed by investigating changes in the combination of March SI6,
September SI6 and September SI12 for a given year. By visualizing the relationship between these metrics, this approach goes
beyond assessing changes in seasonal and annual SI independently (e.g. Spinoni et al., 2018; Vicente-Serrano et al., 2021)





in making use of the multiscalar property of these indices. With an accumulation period of 3 months, Spinoni et al. (2018)
found decreasing occurrence of drought events in winter and increasing occurrence in the other seasons, with the strongest
increases in summer, with a spatial pattern dependent on the scenario and drought intensity considered. These and our results
are in disagreement with Rahiz and New (2013), who found larger and more widespread drought frequency and intensity in the
hydrological wet season. Differences between the drought indicator and the simulations may contribute to this, but likely the
difference mostly stems from a difference in the delineation of seasons. Whereas this study and others use the 3- or 6-month
aggregated drought indicator for the last month of a season to represent that season, Rahiz and New (2013) used the 6-month
aggregated drought indicators for all months of that season.

Changes in drought and extreme drought extent (Fig. 6 and Fig. 7), frequency and regional contrasts (not shown) are ex-
acerbated when a longer timescale is used. Our results and Hanlon et al. (2021) both find that drought changes using longer
aggregation periods increase in magnitude and in spatial contrast.

## 6.2   Differences between SPI and SPEI projections

We show that the magnitude of the difference in projected change between SPI and SPEI is substantial. Using the 6 month
aggregation period, it is comparable to the difference between +2 °C and +4 °C of warming above pre-industrial levels for
the extent and frequency of drought and extreme drought. Within both warming scenarios, the difference in GB-averaged
projected total drought frequency between SPI and SPEI is similar to the ensemble range, for either SI. For extreme drought,
the difference between SPI6 and SPEI6 is similar in size to the ensemble range according to SPI at +2 °C, and lies between the
ensemble ranges of SPI and SPEI at +4 °C.

Previous studies found divergence in trends of drought characteristics between SPI and SPEI in observations (Stagge et al.,
2017; Karimi et al., 2020; Ionita and Nagavciuc, 2021), historical climate simulations (Chiang et al., 2021) and future climate
projections (e.g. Arnell and Freeman, 2021; García-Valdecasas Ojeda et al., 2021; Wang et al., 2021; Ogunrinde et al., 2021),
with SPEI indicating increased drying compared to SPI. Increases in PET under a changing climate, combined with the high
sensitivity of SPEI to PET changes, cause amplified projections of climatological drying and even a reversal of wetting trends
in some parts of the world compared to when only changes in precipitation are considered (Cook et al., 2014). For the UK,
Arnell and Freeman (2021) found that projected increases in drought frequency based on SPEI6 exceeded those based on SPI3,
which is attributed to the inclusion of the effect of PET in SPEI, although the aggregation period difference likely amplified
this effect. By applying the delta method using UKCP18 probabilistic projections, Arnell and Freeman (2021) found that in
a warming climate the frequency of severe drought (SI < -1.5, McKee et al. (1993)) based on SPI3 and SPEI6 increases in
England under both scenarios, are likely to increase under RCP8.5 in Wales and Northern Ireland, and are projected to undergo
little change in Scotland. For the Iberian peninsula, (García-Valdecasas Ojeda et al., 2021) found greater projected increases
in average drought intensity, frequency and duration using SPEI than SPI. Ionita and Nagavciuc (2021) compared trends in
SPI, SPEI and the self-calibrating PDSI from 1901 to 2019. They found divergence of SPI and SPEI trends, especially for
the Mediterranean and Central Europe regions (but also in East and South East England, which are in their Northern Europe
region). They found that the trends in moderate, severe and extreme drought frequencies based on SPEI and SPI showed trends





with opposite signs (increasing for SPEI, decreasing for SPI). Stagge et al. (2017) found that the difference between SPI and SPEI in the area fraction of European land area in drought increased with 2.8% per decade on average from 1980 to 2014.

The observed divergence was driven primarily by decreasing SPI-based drought area not reflected in SPEI-based drought area trends.

## 6.3 The role of AED in drought processes

The greater increases in drought frequency, extent and maximum durations for SPEI-based drought suggest that AED is an important driver of drought changes under global warming. When looking at SPI and SPEI as proxies for the surface water

balance, the assumptions are respectively that no actual evapotranspiration (AET) occurs, or that AET always occurs at its maximum rate (AED), neglecting possible limitations from moisture supply. Neither of those assumptions would hold all of the time. An extensive review of the role of atmospheric moisture demand in drought processes can be found in Vicente-Serrano et al. (2020b). The role of AED (which in the SPEI is characterized by PET) in drought processes, and thus how PET increases translate to changes in drought characteristics, is complex (Vicente-Serrano et al., 2020b), and the SPEI drought index should

be interpreted in the context of the regional climate. Globally, Vicente-Serrano et al. (2013) found stronger correlations of SPEI with vegetation growth in more arid biomes with more negative annual water balances (in regions where cold temperatures are not limiting evapotranspiration). High AED can be expected to increase soil moisture deficits mainly where long-term AED exceeds long-term precipitation, as well as during periods of low precipitation in humid areas (Vicente-Serrano et al., 2020b). Tomas-Burguera et al. (2020) found that the SPEI was more sensitive to AED during periods of low precipitation in humid

areas, as well as in dry to subhumid regions. They argue that overestimation of drought severity due to increasing AED would therefore be very limited and that SPEI is a robust drought indicator.

PET is projected to increase as the climate continues to warm (Fig. 2), and observation-based PET increases for GB were identified by Kay et al. (2013) and affirmed by Blyth et al. (2019). On an annual basis, AET in GB is generally water-limited toward the East and energy-limited in the North and West (Kay et al., 2013). Regional differences in the relative strengths

of AET correlation with precipitation and radiation follow a slightly different pattern, with GB positioned in the transition between humid, radiation-controlled Northern Europe and more arid, precipitation-controlled Southern Europe (Teuling et al., 2009). The negative correlation between annual streamflow and AED also varies regionally, with the strongest correlations (<-0.4) found in North England, the East Midlands and a small area in Scotland, and is weaker than the positive correlation with precipitation (Vicente-Serrano et al., 2019). Based on MORECS data, Kay et al. (2013) found that observed trends in PET

between 1961 and 2012 are greater than those for AET for England and Wales (1.0 mm year$^{-1}$ year$^{-1}$ vs. 0.7 mm year$^{-1}$ year$^{-1}$), while in energy-limited Scotland PET and AET trends are very similar (0.6 mm year$^{-1}$ year$^{-1}$). This is contrasted by a later study by (Blyth et al., 2019) which used the JULES land surface model, who found that, due to increases in precipitation and the large contribution of interception to total AET, the modelled AET increased at a greater rate than PET in GB between 1961 and 2015 (0.87 +- 0.55 mm year$^{-1}$ year$^{-1}$ vs 0.74 +- 0.66). The findings from these studies for GB suggest that future changes

in soil moisture and hydrological drought may not necessarily follow the same regional patterns as the SPI and SPEI-based drought changes, and that for some regions the SPEI may give more of an estimation of drought risk than for others.



There are several arguments indicating the potential of SPEI to overestimate drought risk. When moisture availability is not limiting, increased PET can lead to increased evapotranspiration and thus result in loss from open water bodies and soil moisture (which impacts water bodies by decreasing intermediate moisture fluxes). In humid regions, increasing AED directly

intensifies hydrological drought by increasing the actual evaporation from rivers, lakes and reservoirs, while in drier regions limiting surface moisture availability reduces the drying effect of higher PET on soil moisture (Manning et al., 2018; Vicente-Serrano et al., 2020b). Similarly, during extreme droughts, the effect of increased PET on actual evapotranspiration is reduced as soil moisture becomes the limiting factor and transpiration is reduced due to plant wilting (Van Loon, 2015). Berg et al. (2017) pointed out that projections relying on offline measures of aridity based on a climatic water balance better represents

average dryness trends of the soil surface compared to deeper soil layers and the total column. This implies that the depth distribution from which plant roots extract water is an important parameter determining how SPEI dryness affects vegetation. One of the main criticisms of the SPEI is the divergence between PET and AET under moisture-limited conditions (e.g. Manning et al., 2018). This rests on the assumption that increasing ET, following from high PET, would be indicative of a drought condition, which Berg and Sheffield (2018) pointed out to be a paradoxical line of reasoning since increasing ET

would imply sufficient moisture availability, which under drought conditions (which SPEI aims to quantify) may not be true. Berg and Sheffield (2018) warn that land-atmosphere feedbacks might be "double-counted", as drought conditions raise PET themselves through increasing sensible heat and decreasing humidity. However, as the UK has a strongly maritime influenced climate, these feedbacks may be comparatively less important here Rowell and Jones (2006). Another criticism of the SPEI has to do with the aggregation periods: it is inherently implied that the drying caused by low precipitation and high PET anomalies

take place over the same time scale - which is not the case (Manning et al., 2018), for example in a scenario where a hot period of high PET accelerates a rapid drought onset. The method used to compute PET can introduce severe overestimation of drought risk. This is not only true for temperature-only methods such as Thornthwaite, but also for the Penman-Monteith method, which is widely used to quantify PET based on climate model outputs, including in this study (Milly and Dunne, 2016). Increasing $CO_2$ levels lead vegetation to decrease stomatal conductivity, which leads to less transpired water per unit of

carbon assimilated (). The combined effect of this and increasing vegetation growth due to $CO_2$ fertilisation is expected to be a bulk reduction in transpiration. Usage of reference crop Penman-Monteith PET almost always assumes a constant stomatal resistance. This omission has been identified as an important source of off-line PET overestimation in climate change studies (Milly and Dunne, 2016), although Scheff et al. (2021) show that the $CO_2$ effect on vegetation only explains a small part of the gap between dryness indicators that combine P and PET and impact variables simulated by the climate models themselves

(runoff, runoff ratio, deep-layer soil moisture), and don't fully close the gap between these indicators and vegetation-related variables (leaf area index, gross primary production). The impact of the representation of $CO_2$ and temperature influences on transpiration is likely relevant for the results presented here, as transpiration and bare soil evaporation respectively make up the largest and smallest fractions of total AET in all GB regions, and no evapotranspiration process is negligible in any subregion (Blyth et al., 2019). However, it may be more important toward the south east: whereas in Scotland there is almost as much

interception as transpiration, in the English lowlands interception and bare soil evaporation make up similar shares, which combined form a smaller flux than transpiration alone (Blyth et al., 2019).



To summarise, the SPEI-based results in this work are alarming, but should be interpreted as a conservative upper limit of future atmospheric drought risk due to the arguments presented above. These results do highlight the importance of understanding (changes in) the role AED plays in GB droughts and overall hydroclimate under a changing climate.

## 6.4    From atmospheric indicators to agricultural, hydrological and ecological droughts

The relationship between projected future changes in SPI and SPEI-based drought and changes in soil moisture or hydrological drought varies regionally, as shown in global (e.g. Touma et al., 2015; Vicente-Serrano et al., 2020a) as well as regional (e.g. Lee et al., 2019), Meresa et al. (2016) studies. Many studies have used a range of drought impact data to investigate the relationships of SI with different aggregation periods in GB (Bachmair et al., 2016, 2018; Parsons et al., 2019) and beyond (Gampe et al., 2021). This is not straightforward, as impact metrics of past droughts (based for instance on observed flow) reflect more than the consequences of atmospheric moisture deficit. They are also influenced by water fluxes driven by the land surface (e.g. evaporation limited by soil moisture) and humans (e.g. irrigation and water abstractions), which are not accounted for by SPI or SPEI. Additionally, previously established relationships between drought indicators and impacts may change under a changing climate. Feng et al. (2017) showed a decreasing correlation between simulated SPEI-based drought and soil moisture for the Great Plains in the United States, for aggregation periods of 1 and 12 months and using surface and total column soil moisture.

Several studies have attempted to draw relationships between SI at different aggregation periods and agricultural drought impacts in the UK, using reported impacts (Stagge et al., 2015a; Bachmair et al., 2016; Parsons et al., 2019), crop model outputs (Haro-Monteagudo et al., 2018) and remote sensing products (Bachmair et al., 2018) to measure impacts. Based on these studies, SPI and SPEI (with any aggregation period) perform similarly well for predicting agricultural impacts in the UK, although (Parsons et al., 2019) found SPEI to consistently slightly outperforms SPI at all aggregation periods for lumped reported UK impacts, while (Bachmair et al., 2018) found SPI to be best linked to remotely sensed crop health for most of the UK. Using different datasets of reported agricultural impacts, the indices that best predict total UK impacts were SPEI6 in (Parsons et al., 2019) and the interaction term/joint influence of SPI12 and SPEI12 in (Stagge et al., 2015a), who pointed out that the impacts they studied were dominated by irrigated potato crops and were for a large portion located in regions with productive aquifers, which would explain the longer aggregation period. Parsons et al. (2019) and Bachmair et al. (2016) agree that using SPI or SPEI made little difference regarding which aggregation periods were most linked to impacts. The response of remotely sensed crop health to SI includes large regional variations in the best performing aggregation period, with SPI3 (and SPI2) performing best in East and South England, SI12 performing best in the East Midlands and Yorkshire and Humber, SI1 performing well in Scotland and generally a mix of SI1-4 elsewhere (Bachmair et al., 2018). This agrees with Haro-Monteagudo et al. (2018) who found that the SI3 showed the strongest correlation with modelled crop yield for the East of England. This preference for shorter aggregation periods contrasts with the superior performance of aggregation periods between 7 and 12 months (for either SI) in Yorkshire and Humber, Central England, South East and East England based on reported indicators found by (Bachmair et al., 2016). These large geographical differences in preferred SIn, combined with geographically varying responses to the same SI-based drought intensity (Parsons et al., 2019), suggest that relying on




a single SI with one aggregation period as the indicator of agricultural drought risk in GB (or a similarly diverse region) may be insufficient. Finally, seasonality is an important factor in determining agricultural impacts from drought, as impacts and indicator-impact relationships vary with the growth season. Reported and modelled agricultural drought impacts show the strongest response to SI with aggregation periods of 1 to 6 months in July and August (Haro-Monteagudo et al., 2018; Parsons et al., 2019), while Stagge et al. (2015a) found the highest probability of impacts in September. The large projected increase in summer droughts can thus be expected to have a major impact on agriculture in GB.

Generally, linking SI to hydrological drought and impacts on water resources has received less attention in literature than agricultural impacts. Bachmair et al. (2016) found that agricultural impacts showed strong links to shorter aggregation periods than hydrological impacts (7-8 months vs. 12-24 months) for the subset of regions where this comparison was possible Using the standardized streamflow index, Barker et al. (2016) found that hydrological drought showed stronger correlation with longer SPI aggregation periods (up to 16 and 19 months) for catchments towards the South East of the UK, somewhat in contradiction to the SPI2-6 (best: SPI3) identified as best predictors by (Folland et al., 2015) using a similar method. In the North and West short SPI aggregation periods (SPI1-6) were generally best correlated with streamflow drought, with SPI1 being the most common best predictor for streamflow drought among the studied catchments. By comparison, the SI aggregation periods best predicting reported hydrological drought and water supply impacts according to Bachmair et al. (2016) are longer, while retaining the same spatial pattern. They found SPI24 and SPEI24 the best predictors of hydrological drought and water supply impacts in the West Midlands and the South and East of England, but aggregation periods from 4 to 8 months in Yorkshire and Humber. In North West England, the aggregation periods and best indicator varied from SPI(3-)4 to SPEI8 depending on the model for reported hydrological drought and water supply impacts, while in Wales depending on the model SPEI8 or SI7-8 came out as best predictor. (Folland et al., 2015) found that SPI8-14 (best: SPI12) were highly correlated with the standardized groundwater index, a groundwater drought indicator. Two studies linking SI to hydrological drought and water resources impacts compared SPI and SPEI. Stagge et al. (2015a) found a combination of SPEI3, 9 and 24 as the best predictor for water supply impacts, while in Bachmair et al. (2016) SPEI had a very slight advantage over SPI in some regions. Projected changes in streamflow drought tend to lie in between projections for SPI and SPEI, but can also lie outside of this range. In Touma et al. (2015), changes in exceptional streamflow drought were closer to the SPI- than the SPEI-based projections in most regions of the world, with more resemblance to the SPEI-based changes being found in more Northern latitudes. For a catchment in South Korea, Lee et al. (2019) showed that the magnitude of changes projected for hydrological drought intensity and frequency were in between the projections for SPI and SPEI (with the latter consistently projecting large increases).

As for ecological droughts, remote sensing-based forest health indicators were most highly correlated with SPI3 in the East and South East of England and SPEI1 in the rest of the UK by Bachmair et al. (2018) (with local variations and stronger links to higher aggregation periods in some regions). The presence of freshwater ecosystem impacts was linked to similar aggregation periods as the hydrological drought and water resources impacts by Bachmair et al. (2016), with SI24 the best indicator in the East and South East of England and the West Midlands, SPEI24 in Wales, and SI6 to SI24 depending on the model used in other regions.





To summarise, while studies linking SI to impacts agree in some aspects (e.g. longer SI aggregation periods for predicting streamflow drought in the south east than the north west), there is a lot of uncertainty left. In the UK, due to regional differences in climatology, hydrogeology and agricultural practice, the links between SI and various impacts are more meaningful at regional or local levels than at the national scale (e.g. Barker et al., 2016; Parsons et al., 2019). Socio-economic and physical vulnerability factors also influence the impacts resulting from droughts characterized by certain SPEI or SPI values (Blauhut

et al., 2016). Additionally, previously established relationships between drought indicators and impacts may change under a changing climate Feng et al. (2017). It is therefore difficult to quantitatively infer changes in agricultural, ecological and hydrological drought from projections of SPI and SPEI alone. Based on the literature discussed above and our results, agricultural drought impacts may be expected to increase due to projected increase in summer drought frequency and intensity (Fig. 8 and Fig. 9), which is found for both indicators in most of GB, including in agriculturally important regions. However, as the

projections based on SPI are much milder than those based on SPEI, magnitude of this increase depends on the importance of increasing AED and temperature for root zone soil moisture and crop growth. The greater frequency and intensity of dry years (SI12), as well as the increasing extent and frequency of drought and extreme drought with longer aggregation periods, may indicate greatly increased risk of drought impacts on water resources in the southeast and east, and by extension irrigated agriculture in these regions. Smaller projected increases in drought frequency based on SI3 may indicate similarly smaller

increases in streamflow drought in the northwest. Based on studies comparing rainfall-only and PET-including indicators to runoff in climate models themselves (), the SPEI drought projections may give a large overestimation for streamflow drought, and instead changes in streamflow drought might lie in between projections for SPI and SPEI, depending on the effect of increasing AED and temperature.

## 6.5   Study limitations

The set of regional climate projections in UKCP18, which this study relies upon, is not intended to represent a comprehensive, probabilistic view of possible changes, but rather to sample a broad range of possible futures and provide storylines suited for analysis of impacts (Murphy et al., 2018). The UKCP18-RCM projections were produced using the same GCM and RCM structure with perturbed parameter values, meaning that the climate model structural uncertainty has not been sampled. Finally, as opposed to an ensemble where only the initial conditions differ, the projections of such a perturbed physics ensemble cannot

be combined in order to obtain longer time series for each level of global warming. This especially limits our analysis of multi-year droughts, which have a lower probability of occurrence and are influenced by interdecadal variability. For those regions, the length of the time slices used is also a limiting factor for investigating projected changes in the occurrence of such events.

    The drought indices this study uses are among the most widely used ones in academia and operational drought management. However, other indices exist that rely on precipitation or some combination of precipitation and AED. Choosing a different

drought index that includes both moisture supply and demand, with a different degree of sensitivity to each component, could lead to slightly different results (Vicente-Serrano et al., 2015). The drought index choice itself is a source of uncertainty, as highlighted by this study and previous studies.





Another limitation of the study lies in the vegetation assumptions made when calculating PET using Penman-Monteith for the FAO56 reference crop. Transpiration is responsible for the largest fraction of total evapotranspiration in the UK (Martens et al., 2017; Blyth et al., 2019), and so any assumptions affecting this may have important effects on the actual changes in PET and AET, and thus how representative SPEI is of the climatic moisture balance anomaly. Vegetation assumptions in the UKCP18-RCM projections themselves present another important limitation. In the UKCP18 "Soil Moisture and the Water Balance" fact sheet, Pirret et al. (2020) write that "the models use prescribed vegetation, which means that the model does not represent how increasing atmospheric carbon or reduced soil moisture would affect vegetation, or any feedbacks that this may have on the atmosphere or land surface". For reasons discussed in section 6.3, this may lead to unrealistic changes in AET under a warming atmosphere with increasing $CO_2$, and, thus introduce errors in the simulated temperature and humidity, which in turn affect PET.

## 7 Conclusions

We used the regional climate model perturbed parameter ensemble from the latest set of national climate projections for the UK, UKCP18, to quantify projected changes in drought characteristics. For this, two atmosphere-based standardized drought indices were used that attempt to capture different processes: the Standardized Precipitation Index (SPI) and the Standardized Precipitation Evapotranspiration Index (SPEI). The SPI gives the anomaly of n-month aggregated precipitation, expressed in standard deviations from the mean n-month aggregated precipitation in a calibration period. The SPEI is similar, except the variable being standardized is a climatological moisture balance given by precipitation minus potential evapotranspiration. We assess in detail the difference between these indices for investigating the impact of climate change on drought frequency, extent, seasonality and duration, for two categories of drought intensity. This is the first detailed systematic analysis of SPI- and SPEI-based drought projections and their differences for Great Britain.

Drought risk over Great Britain increases almost everywhere with increasing global mean surface temperature, including extreme drought risk. We find projected increases in drought frequency, extent and intensity (assessed through the increase of extreme droughts) with global warming. The projected changes in drought frequency, seasonality and duration show large regional differences across GB, with the greatest increases generally found in English regions and Wales, and little change (or even decreases) drought in North and West Scotland. Droughts of all extents are projected to increase, including events more widespread than the maximum extent in the observations and reference period simulations. Unsurprisingly, increasing summer droughts are the main contributor of increasing frequency of increasing longer-term dry conditions. Contrasting years that consist of a wet winter combined with a dry summer are also projected to increase in occurrence, however the combined result of contrasting seasonal changes is a projected increase in dry years for most regions. Changes in the distribution of drought event durations are projected. For both indicators, but especially for the SPI, the changes are far greater by +4 °C than by +2 °C, supporting the general consensus that every additional degree translates into increasing extreme events.

The choice of atmosphere-based drought indicator can have a great impact on the derived drought characteristics, and thus great care should be taken when selecting a drought index for climate change studies. While using the SPI the UKCP18-RCM





ensemble projects some increase in drought frequency and extent, these changes are far greater when using SPEI. The difference between the 6 month aggregation period based indicators is similar in magnitude to the ensemble range of GB-averaged total and extreme drought frequency, and the +2 °C SPEI projections better resemble the SPI-based projections under +4 °C than under +2 °C for drought and extreme drought frequency, spatial extent and seasonality. The spatial pattern of simulated drought

frequency is similar between the indicators, but there are subtle differences. Projected changes in the distribution of drought durations also differ between the indicators. Droughts shorter than 6 months are projected to increase in occurrence in most regions based on the SPI, but projected to decrease based on the SPEI in many of these regions. On the other end, the occurrence of multi-year droughts lasting over 3 years (based on 6-month aggregated indicators) is only projected to increase using the SPEI.

With the sizeable divide between projections based on both indicators, it becomes increasingly important to understand how atmospheric evaporative demand and temperature affect droughts and their propagation to impacts in GB. The large difference between SPI and SPEI in our results calls attention to the need to understand the influence of AED changes on GB drought, and the importance of its simulation. Different modelling approaches can help understand future changes to the impact of atmospheric moisture demand, as well as changes to the supply side of the surface water balance. As part of this,

the simulated soil moisture, evaporation and runoff calculated in the UKCP18-RCM itself (Pirret et al., 2020) can be used for future analysis of UK drought risk and to complement this work. Moreover, both land surface modelling and hydrological modelling approaches are valuable to shed light on projected changes in different components of the hydrological system. More generally, this work raises the question of how these changing drought characteristics translate into impacts for agriculture, water resources and ecosystems in GB. As SI are used as proxies for different types of drought, it is valuable to understand how

established relationships between these indicators and impacts hold up under a changing climate. Under the current climate, according to the reviewed literature there is little difference between SPI and SPEI in their ability to predict different drought impacts. However, this is likely to change as SPI and SPEI diverge due to increasing PET. To this end, a comparison of outcomes of impact simulations with these SI and similar drought indicators may help guide indicator applications in practice.


*Code and data availability.*

SPEI and SPI data are available on Zenodo (doi:10.5281/zenodo.6123020) (Reyniers et al., 2022b). Bias adjusted UKCP18-based PET is available on Zenodo (doi:10.5281/zenodo.6320707) (Reyniers et al., 2022a). The research and visualisations was carried out in Python. Python code for computation and analysis is available upon reasonable request.






*Author contributions.*

All co-authors were involved in designing the study. NR carried out the research. NR wrote the manuscript and designed the
visualizations, with input from TJO and NA. All co-authors provided helpful feedback to the manuscript and approved of its
final version.

*Competing interests.*

The authors declare that they have no conflict of interest.

*Acknowledgements.* NR is funded on a 50/50 basis by Anglian Water Ltd. and University of East Anglia. The authors would also like to
acknowledge the data made available by the Met Office (UKCP18, Had-UK Grid) and CEH (CHESS-PE).



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
