# Peer review of "Projected changes in droughts and extreme droughts in Great Britain strongly influenced by the choice of drought index"

_Hydrology and Earth System Sciences, 2022_

## Referee Comment (RC1)

Review - Projected changes in droughts and extreme droughts in Great Britain strongly influenced by the choice of drought index

This paper examines projected changes in drought frequency, extent, seasonality and duration for Great Britain using the latest UKCP18 projections. It analyses the differences between two atmospheric drought indicators (SPI and SPEI) and shows that the choice of drought indicator can have a big impact on the derived drought indicator.

Overall, this is an interesting study that is well written and well presented. The analyses are extensive, well thought out and well executed. I believe the paper will have appeal to a wide readership, however, there are some core issues that need to be resolved to enhance the key messages of the paper. Firstly I think the motivation of the study needs to be more clearly defined and more detail is required on the choice of climate projections and bias correction. Secondly, I strongly encourage the authors to take a critical read of the discussion and shorten it to the core messages – it is currently very long and your interesting results are getting lost.

More detailed comments are provided below that I hope the authors find useful.

**Comments**

1. **Motivation.** The motivation for the study needs to be clearer. Currently the introduction reads like a series of definitions rather than a compelling story of why this study should be undertaken. There are two core areas where I think this could be improved:
    a. There have been quite a few studies that have used or compared drought indices (many are cited in your discussion) so what does this study offer that is novel and different?
    b. Why focus on Great Britain (which often isn't thought of as a country that experiences lots of droughts!) and how are your research questions relevant to this region?
2. **Use of UKCP18.** There needs to be better justification for the use of the regional projections from UKCP18 in this study – why use this set of projections instead of the local UKCP18 projections for example? Or use a set of climate projections that encompasses different GCMs or RCP scenarios (for example)?
3. **Bias Correction.** At the end of Section 2.2 there is a section on bias correction which needs more detail. Why did you choose these bias correction methods and how are they applicable to the types of biases you observe between the climate projections and observational data? It would be helpful to add some plots in the supplementary information showing these biases to help the reader understand the nature of the biases and how they were corrected. For example, you note in section 3.2 that a direct comparison of the results between climate model ensemble members and observations is only possible because their distributions are similar – it would be helpful to see evidence of this.
4. **Discussion.** The discussion section is extremely long and as a result, a lot of your interesting results get a little lost amid all the discussion. The authors need to have a critical read of the discussion and carefully consider if all the text is needed. As a suggestion, I would significantly shorten section 6.4 as this tends to be a review of the

literature, rather than a discussion of your results with context from the literature (you could just use the summary paragraph – you don't really need all the preceding text).

**Minor/Technical Corrections**

P2 L45. 'Drought indices that only rely on atmospheric data are a popular choice due to data availability and **propagating model uncertainties**.' I don't understand this sentence – why are they a popular choice due to propagating model uncertainties?

P3 L65. You could add into the third research question the uncertainty in the RCM as a lot of your results focus on the differences between ensemble members. E.g. How sensitive are the projected changes in drought characteristics to the choice of atmosphere-based drought indicator and parametric uncertainty in regional climate models?

P3 Section 2.1. It may be useful to state the time-period you used from each observational dataset.

P5 L141-142. Why did you include aridity as well? What does it add to the story? The motivation could be a little clearer.

P6 L168. The area for London is small, but it is a central hub which contains around 14% of the population of GB and is likely to be significantly affected by droughts in the future. Consequently, leaving out the figures for London because the area is small seems an odd choice, given the significant impacts changes in droughts will have in this small part of GB. Potentially a better reason would be because the results are very similar the South-East region or East of England?

Figure 1. It is difficult to see the labels for North and West Scotland – can these be moved or made clearer?

P9 L203-205. This sentence is a little difficult to understand – can it be rewritten?

P9 L209. 'For extreme meteorological drought, all ensemble members **project multiples** of the reference period frequency by +4 $\circ$C.' I don't understand what you mean here.

P10 L233 'due to SPEI6 occurrences beginning to saturate when they have already become quite frequent'. What do you mean by 'beginning to saturate'?

P28 L666. I think you are missing some key references from the brackets?

Code and Data Availability – Great to see that the data you produced are available but this section needs to be a full description of all the data used in the study, including links to all the data you used for analysis (i.e. for the CHESS-PE, HadUK and UKCP18 data). I would also reword to 'The SPEI and SPI data **produced in this study** are available on Zenodo (doi:10.5281/zenodo.6123020) (Reyniers et al., 2022b) alongside the bias adjusted UKCP18-based PET (doi:10.5281/zenodo.6320707) (Reyniers et al., 2022a).'

---

## Referee Comment (RC2)

**Review Comment hess-2022-94**

**Title: Projected changes in droughts and extreme droughts in Great Britain strongly influenced by the choice of drought index**

In this paper changes in drought characteristics are evaluated for GB, for 2 future climate scenarios. Two drought indices are used to characterize drought severity, SPI and SPEI, for various space and time-scales. The study finds increase in most drought characteristics (frequency, extent, duration etc) for future climate conditions, not entirely unexpected.
In particular, the authors emphasize that the choice of drought index influences the quantitative assessments of projected drought changes.

Given this perspective, it is particularly important to document not only the indices used but also the full range of methods applied to reach their conclusions. This is the main problem I see with this paper: many aspects of the methods used for analysis are not clearly explained.
Secondly, the Results and Discussion sections are very long and need to be strongly condensed to convey only the essential information. To give an example, section 5.1 covers almost 2 pages to describe a single figure, followed by 2 more pages for a second figure. That's a lot of descriptive information that can be drastically shortened, based on a critical reassessment of what pieces of information are really important and worth for the reader to know.

**Detailed comments:**
Abstract:
- General comment: the summary of results presented here is quite superficial, i.e descriptive rather than interpretive. Deeper interpretation of the results would make the Abstract a lot more appealing.
- Check phrasing here: the phrasing suggests that projected changes are sensitive to the choice of drought index (L5). However, projected changes are simply what the simulated climate scenarios tell us, how they are summarized in quantitative metrics is where the differences in interpretation come in.
  Same confusing phrasing is used throughout, e.g. (L14) "SPEI results in greater increases in drought frequency and extent". Obviously the drought characteristics do not change, only how the indices are computed. L16: "projected changes (..) depend on the drought index, (..)". Again, reasoning is flawed: projected changes are the same, the indices are different, not the other way around.

1. Introduction:
- P2, L 39: it is suggested here that evapotranspiration only depends on atmospheric variables, but strictly speaking vegetation also plays a role (stomatal conductance)
- P3, l65: same phrasing issue as in Abstract

3. Methods:
- P5: it would be helpful to provide the definitions (and/or the equations) of the indices that are used in the paper (AI, SPI, SPEI), so the reader doesn't need to search back in the literature
- P5, L151: "observation-based calibration": this needs clarification. How was this calibration done, this is currently not explained.
- P6, section 3.3, Drought characterization: it is stated that spatiotemporal characterization is important - agreed. Unfortunately, the authors do not specify the space and time scales used in their characterization. What is "regional", "seasonal", what range of space and time scales did they investigate?
- P6, L162: please clarify definition of 'extreme drought'. At present, the choice of SI<-2 sounds arbitrary
- P6, L177: "a distribution fitted to the relatively short times series". This needs explanation: what distributions were fitted, how exactly?

4. Projected climate changes:
- In the caption of Figure 3 it is mentioned that "after bias adjustment using change preserving quantile mapping" is applied to the ensemble members.
  This is not the right place to mention such a data processing step! Please explain adequately in the main text.

5. Projected changes in drought characteristics:
- L204: the authors refer to "2C above pre-industrial", but as far as I understand their reference scenario is 1981-2005. That's not exactly pre-industrial.. Please clarify or correct.
- Figure 4: the use of % as a unit for frequency is very confusing here. If I understand correctly the % is calculated based on number of years (in 25 year climate period) that index values are below a given threshold. This is a guess, it is not clearly explained.
  Much later, in Figure 10, the authors use "number of events" instead -  a much more straightforward type of unit. I recommend using this unit throughout.
- LL 199-241: this is a very extensive description of a single figure (see earlier comment). Please reflect critically: what pieces of information are really worth mentioning?
- LL 242-290: same here, figure description is far too lengthy.
- L246: "the fit of the gamma and GEV distributions used in the calculation of SPI and SPEI".
  So gamma and GEV distributions were fitted apparently..? This should have been explained in the Methods Section!
- L266: "detrended temperature simulations". Again, please explain this properly in the Methods section – how was the detrending done, for what purpose exactly?
- L272: "purely temperature-based PET" : this seems to suggest that temperature has a strong influence on PET, yet the influence of Radiation is much stronger (linear relationship with PET in Penman equation). Please check the reasoning here, it seems flawed.
- Figure 6: this is first time Observations are shown in any of the results graphs! Why only now and not in the earlier graphs?
  Also in Figure 6: a gradual color scale is applied here which makes it impossible to distinguish clearly between the 3 scenarios. Note that in the current representation their seems to be no significant difference between the Reference and +2C scenario.

Note: I stopped reading here. Sections 5 and 6 are very lengthy and many of the results point in the same direction. Are all these figures and subsections really needed to make the point stated in the title, that "Projected changes in droughts are strongly influenced by the choice of drought index"?
I strongly recommend that the authors take a critical view of their results and make a selection of the materials that most strongly support their conclusions. Then report these clearly and concisely.

---

## Author Comment (AC1)

**Response to RC1 (anonymous)**

Thank you for your comments, we are glad you find the study interesting and appealing to a wide readership. We generally agree with your comments, including the main points that the motivation for the study should be better explained and that the discussion would benefit from significant shortening to focus on the core information. We are happy to implement the requested or suggested changes to the benefit of the paper.

Please find below our responses to the individual comments.

**Comments**

1. **Motivation.** The motivation for the study needs to be clearer. Currently the introduction reads like a series of definitions rather than a compelling story of why this study should be undertaken. There are two core areas where I think this could be improved:
   1. There have been quite a few studies that have used or compared drought indices (many are cited in your discussion) so what does this study offer that is novel and different?
   2. Why focus on Great Britain (which often isn't thought of as a country that experiences lots of droughts!) and how are your research questions relevant to this region?

   We will take care to expand on these two core areas when writing a more compelling introduction. On 1.: indeed, many studies have compared drought indices, but we go in more depth here than is usually done, considering the differences in a wider range of drought characteristics. On 2.: First, the hydroclimatology of Great Britain is generally humid but still quite diverse, and drought is already a concern – especially in the South and East – as has also been shown in recent years. Moreover, water managers across Great Britain are facing challenges from different angles to secure adequate water supply in the future, including increases in demand and indeed climate change impacts. As this is a humid region, the primary concern with regards to future drought resilience is typically precipitation. This paper invites a critical examination of the potential importance of evaporation in a drier Great Britain due to climatic changes.
   Moreover, it points out just how different the resulting changes in a range of statistics are based on which atmospheric-based drought indicator was chosen to quantify droughts – arguably not surprising, but nevertheless a finding that is often underappreciated.

2. **Use of UKCP18.** There needs to be better justification for the use of the regional projections from UKCP18 in this study – why use this set of projections instead of the local UKCP18 projections for example? Or use a set of climate projections that encompasses different GCMs or RCP scenarios (for example)?
   We will improve the justification for the choice to use the UKC18 regional projections in our revised manuscript, along the following lines. The UKCP18 simulations are the de facto national projections for the UK, and have been produced with the aim of providing a spread of projections to support adaptation efforts in the UK. The local projections would have provided a greater added value if intense (convective) precipitation events on subdaily time scales or the added spatial resolution would have been crucial, however as droughts tend to be more spread out in space and time, we judged that the 12km daily resolution of the UKCP18 RCM pose a better trade-off between practicality and spatiotemporal detail for this purpose. While we agree that an ensemble that encompasses multiple GCM-RCM structures would add

another interesting dimension to the study, expanding the ensemble with e.g. EURO-CORDEX simulations would have been outside the scope and capacity of the study. The UCKP18 regional ensemble already samples parameter uncertainty in both the GCM and RCM, while our use of specific warming levels avoids a focus on particular RCP scenarios.

3. **Bias Correction.** At the end of Section 2.2 there is a section on bias correction which needs more detail. Why did you choose these bias correction methods and how are they applicable to the types of biases you observe between the climate projections and observational data? It would be helpful to add some plots in the supplementary information showing these biases to help the reader understand the nature of the biases and how they were corrected. For example, you note in section 3.2 that a direct comparison of the results between climate model ensemble members and observations is only possible because their distributions are similar – it would be helpful to see evidence of this.

   The focus of this study is not on the bias correction, though we agree that the reader would benefit from information about the model biases. The raw and bias corrected data were evaluated in detail. The results of this evaluation for temperature and precipitation will be presented in a separate paper about bias correction that is currently in preparation (for which the bias adjusted precipitation data will also be made available). We will include in the supplementary information of the current paper plots of the PET and precipitation biases (unless the separate bias correction paper is already available, in which case we will refer the reader to that paper for details of the precipitation biases). Nicole Forstenhäusler has kindly agreed we can include in this response (and the supplementary material of a potential revised version) some of the maps she produced for the bias correction paper.

[Figure]

*Mean precipitation biases in UKCP18-RCM for 1981-2010, expressed as a percentage of the observed values. The bias for each ensemble member was computed and the mean across the ensemble is shown here. Dry-day frequency is the percentage of days with P < 1 mm; q95 is the 0.95 quantile of precipitation. Created by Nicole Forstenhäusler.*

The maps we propose to include for PET will follow a similar approach, showing ensemble averaged biases in the daily PET mean, 5[th] and 95[th] percentiles in the Supplementary.

[Figure]

*Mean PET biases (mm) in UKCP18-RCM for 1981-2010. The bias for each ensemble member was computed and the mean across the ensemble is shown here. Q05 and Q95 are the 0.05 quantile and the 0.95 quantile across.*

The biases we observed for different quantiles were not equal to the biases observed in the mean, so we opted for a bias adjustment method that took this into account. Similarly, biases also varied between months and locations, so the bias adjustment needed to be specific for each month and grid cell. We considered more complex

methods that e.g. take into account different time scales or multivariate distributions, however it was unclear whether these methods would be beneficial compared to the univariate quantile mapping approach that is well-established in the literature. The ISIMIP3b method that was chosen in the end is based on quantile mapping, but also preserves projected changes in the variables being corrected, and adjusts the frequency of dry days separately – a desirable feature for drought research. In addition to the PET (and precipitation) evaluation plots in the Supplementary materials and the paper which will discuss the evaluation of raw and bias corrected precipitation (and temperature), we will also add the observation data to the following figures, so that the reader can see how they compare to the bias adjusted reference period UKCP-RCM simulations: Fig. 2 (aridity), Fig. 3 (average seasonal cycle), Fig. 4 (dots representing the observations on the scatter plots showing SPI6 vs SPEI6 drought frequency).

4. **Discussion.** The discussion section is extremely long and as a result, a lot of your interesting results get a little lost amid all the discussion. The authors need to have a critical read of the discussion and carefully consider if all the text is needed. As a suggestion, I would significantly shorten section 6.4 as this tends to be a review of the literature, rather than a discussion of your results with context from the literature (you could just use the summary paragraph – you don't really need all the preceding text).

We agree that not all the text is needed to support the results and will significantly condense the discussion as such. Thank you for providing an example as a starting point.

**Minor/Technical Corrections**

P2 L45. 'Drought indices that only rely on atmospheric data are a popular choice due to data availability and **propagating model uncertainties**.' I don't understand this sentence – why are they a popular choice due to propagating model uncertainties?

Right, this sentence is skipping a few steps. We meant here that when precipitation and potential evapotranspiration are used to drive hydrological models, we need to make other choices that are uncertain (what (type of) hydrological model, how to calibrate it, …), and so atmospheric-based indicators provide a practical alternative. We will clarify this by replacing the last part of that sentence with "due to their ease of use (they do not require the deployment of an impact model, such as a hydrological model).".

P3 L65. You could add into the third research question the uncertainty in the RCM as a lot of your results focus on the differences between ensemble members. E.g. How sensitive are the projected changes in drought characteristics to the choice of atmosphere-based drought indicator and parametric uncertainty in regional climate models?

Thank you for pointing this out, we will include the element of comparing the sensitivity to the drought indicator choice with the sensitivity to the sampled RCM parameter uncertainty and the GMWL, as this is indeed something we put focus on.

P3 Section 2.1. It may be useful to state the time-period you used from each observational dataset.

We will add this in. From both datasets, we use 1961-2010 for the SI calibration, 1981-2010 for the bias correction and 1981-2005 for comparisons to the reference period in the results.

P5 L141-142. Why did you include aridity as well? What does it add to the story? The motivation could be a little clearer.

It accompanies the seasonal cycle plots of precipitation and potential evaporation in establishing an understanding of the mean climatic changes projected in UKCP18-RCM, before we discuss the drought frequency changes in depth, and providing a metric that is

more intuitive to interpret because it is based on physical rather than standardised quantities.

P6 L168. The area for London is small, but it is a central hub which contains around 14% of the population of GB and is likely to be significantly affected by droughts in the future. Consequently, leaving out the figures for London because the area is small seems an odd choice, given the significant impacts changes in droughts will have in this small part of GB. Potentially a better reason would be because the results are very similar the South-East region or East of England?
In response to a reviewer request to reduce the overall content of the paper, we will reduce some figures in the main text to show only 4 selected regions, with results showing all regions still included in the Supplementary information. We will then take this opportunity to reconsider inclusion of results for London in the latter, if they add relevant different information.

Figure 1. It is difficult to see the labels for North and West Scotland – can these be moved or made clearer?
We will adjust this.

P9 L203-205. This sentence is a little difficult to understand – can it be rewritten?
Agreed, we will adjust this. Proposed rephrasing:
"Using SPI6, drought frequency is projected to increase slightly on average under +2 °C above pre-industrial levels, with larger relative increases for the extremely dry conditions."

P9 L209. 'For extreme meteorological drought, all ensemble members **project multiples** of the reference period frequency by +4 ◦C.' I don't understand what you mean here.
Proposed rephrasing: For extreme meteorological drought, the projected frequency increases between two- and eightfold by +4°C compared to the reference period frequency, across the ensemble.

P10 L233 'due to SPEI6 occurrences beginning to saturate when they have already become quite frequent'. What do you mean by 'beginning to saturate'?
We will rephrase this. The point of this sentence was that, for the SPEI6 and the highest warming level, at some point the (vast) majority of summers are classified as drought, and so summer droughts cannot become much more frequent. As the summer droughts contribute the most to projected increases in drought frequency, the main contribution to the projected drought frequency changes starts to become "saturated" for SPEI.

P28 L666. I think you are missing some key references from the brackets?
Indeed, thank you for spotting, these will be added in.

Code and Data Availability – Great to see that the data you produced are available but this section needs to be a full description of all the data used in the study, including links to all the data you used for analysis (i.e. for the CHESS-PE, HadUK and UKCP18 data). I would also reword to 'The SPEI and SPI data **produced in this study** are available on Zenodo (doi:10.5281/zenodo.6123020) (Reyniers et al., 2022b) alongside the bias adjusted UKCP18- based PET (doi:10.5281/zenodo.6320707) (Reyniers et al., 2022a).'
Thank you very much for pointing this out, we will add this and reword accordingly.

---

## Author Comment (AC2)

**Response to RC2 (Marie-Claire ten Veldhuis)**

Thank you for your review of this study. We generally agree with your comments, are happy to address them, and believe this will substantially improve the paper. In particular, to address a main overarching comment, we intend to significantly shorten the figure descriptions in the Results section as requested, and also reduce the number of panels in some figures, in cases where not all panels are needed to support the information conveyed. On the methodology, we will expand the explanations in response to the comments, but please note that some of the requested information was already present in the Data section (this will be moved to Methods).
Please find below our responses to the individual detailed comments.

Abstract:

- General comment: the summary of results presented here is quite superficial, i.e descriptive rather than interpretive. Deeper interpretation of the results would make the Abstract a lot more appealing.
  We will expand the interpretation of the results in the Abstract.
- Check phrasing here: the phrasing suggests that projected changes are sensitive to the choice of drought index (L5). However, projected changes are simply what the simulated climate scenarios tell us, how they are summarized in quantitative metrics is where the differences in interpretation come in.
  In the IPCC Glossary (IPCC, 2021), "projections" are defined as follows: "*A potential future evolution of a quantity or set of quantities, often computed with the aid of a model. Unlike predictions, projections are conditional on assumptions concerning, for example, future socio-economic and technological developments that may or may not be realized.*".
  Our use of "projections" meets the IPCC definition: in this case, the *model* is a combination of climate models and simple models of drought characteristics. By analogy, if instead of SPI and SPEI we'd estimated drought conditions with two hydrological models, one of which ignored evaporative losses and one of which included them (without making them dependent on moisture availability), it would be valid to say that the future drought projections were sensitive to the choice of hydrological model.
  We propose to address this by being clear and more specific in our wording where this phrasing occurs, e.g. specifying that we mean the projected changes *to drought characteristics.*
- Same confusing phrasing is used throughout, e.g. (L14) "SPEI results in greater increases in drought frequency and extent". Obviously the drought characteristics do not change, only how the indices are computed.
  See previous.
  Proposed rephrasing L14: "In general, far greater increases in drought frequency and extent are found when using SPEI for drought quantification than when using SPI". (Alternatively: replace "results" by "implies").
- L16: "projected changes (..) depend on the drought index, (..)". Again, reasoning is flawed: projected changes are the same, the indices are different, not the other way around.
  See previous.
  Although it is already specified here that this concerns the projected changes *in the distribution of drought durations*, an alternative wording could be: "the quantification of projected changes […] depends on …"

Introduction:

- P2, L 39: it is suggested here that evapotranspiration only depends on atmospheric variables, but strictly speaking vegetation also plays a role (stomatal conductance).
Proposed solution: "[…] reference crop (A 1998), in which a fixed role of vegetation and fixed high moisture availability are assumed, such that only the effect of the atmospheric variables is left in the spatiotemporal variation of the resulting reference evapotranspiration."

- P3, l65: same phrasing issue as in Abstract.
See earlier comment. Alternative phrasing: How sensitive are quantifications of projected changes in drought characteristics to […]"

Methods:

- P5: it would be helpful to provide the definitions (and/or the equations) of the indices that are used in the paper (AI, SPI, SPEI), so the reader doesn't need to search back in the literature.
Thank you for pointing this out. We will add a paragraph and equation to the methods section to explain the standardised indicators (including specific methodological choices such as the fitted distributions used in the SI calculation, see your later comment), so that readers who aren't already very familiar with these indicators aren't required to refer back to the cited literature.

- P5, L151: "observation-based calibration": this needs clarification. How was this calibration done, this is currently not explained.
Proposed wording change:  *[…] was used to fit the distributions for the SPI and SPEI calculation. This observation-based calibration was also applied […].* This, combined with the added SPI/SPEI explanation (see above), will hopefully clarify the statement.

- P6, section 3.3, Drought characterization: it is stated that spatiotemporal characterization is important - agreed. Unfortunately, the authors do not specify the space and time scales used in their characterization. What is "regional", "seasonal", what range of space and time scales did they investigate?
Thank you, we will rewrite this to make it clearer.
The characterisation in space came in 3 forms: (1) each grid cell separately (for frequency, as visualised in maps); (2) UK-averages (extent expressed as fraction of the surface in drought, plus the summary heatmaps); (3) averages by UK administrative regions (for duration of individual events and investigating the seasonal contributions).
The characterisation in time was done in the following ways: (1) frequency of exceeding SI thresholds (% of time); (2) duration and counts of individual events, which are defined as continuously negative SI; (3) seasonal contributions, in which we compared the SI12 for each year to the SI6 representing the October-March and April-September periods making up that year. So in this case, the term "seasonal" is used to indicate hydrological winter (SI6 for March, which represents the anomaly for October-March) or hydrological summer (SI6 for September, which represents the anomaly for April-September).

- P6, L162: please clarify definition of 'extreme drought'. At present, the choice of SI<-2 sounds arbitrary.
This is the threshold used to separate "extreme" drought in the paper that originally proposed the SPI and in many studies that apply standardised indicators of drought. It is also used as the threshold for extreme drought in some drought monitoring

systems (e.g. https://eip.ceh.ac.uk/hydrology/water-resources/). As you rightfully point out, it is a bit arbitrary. A few other studies have used other threshold levels intended to better reflect impacts, however the threshold values linked to other drought types/impacts vary spatially across the UK (as shown by Parsons et al., 2019), so to keep it simple we opted for the thresholds of -2 and -1 standard deviations, to stay consistent with what is conventionally used in much of the literature.
We are happy to add a concise clarification of the "drought" and "extreme drought" threshold choices to the text.

- P6, L177: "a distribution fitted to the relatively short times series". This needs explanation: what distributions were fitted, how exactly?
This will definitely be clarified by the expansion of the SPI/SPEI explanation in Methods (see earlier comment). These distributions are fitted as part of the standardisation process for the SP(E)I calculation.

4. Projected climate changes:

- In the caption of Figure 3 it is mentioned that "after bias adjustment using change preserving quantile mapping" is applied to the ensemble members.
This is not the right place to mention such a data processing step! Please explain adequately in the main text.
It is already explained in the final paragraph of the section on the UKCP18 regional climate projections (L113-118):

*"As comparison to observations revealed significant bias in the simulation of both precipitation and PET, **these variables were statistically post-processed using the ISIMIP3b change preserving bias adjustment method (Lange, 2019) version 2.4.1 (Lange, 2020**). For precipitation, the gamma distribution and mixed additive/multiplicative per-quantile change preservation were used. For PET and PETdtr−tas, the Weibull distribution, detrending and mixed additive/multiplicative per-quantile change preservation were used. A dry threshold of 0.1 mm day-1 was selected below which there is considered to be no precipitation or PET."*

We will move the bias adjustment paragraph to a dedicated subsection in "Methods" instead, and also expand the explanation further in response to a comment from Reviewer 1.

5. Projected changes in drought characteristics:
- L204: the authors refer to "2C above pre-industrial", but as far as I understand their reference scenario is 1981-2005. That's not exactly pre-industrial.. Please clarify or correct.
Thank you for pointing this out, we will clarify this. Indeed, our "reference" period does not represent the pre-industrial level. The UKCP18 Derived Projections report on which we based our time slice selection documents the timings of crossing 2 and 4 °C of warming relative to pre-industrial levels, which refers to the period of 1850-1900 (Gohar et al., 2018). Our "REF" time slice of 1981-2005 was chosen as the UKCP18 RCM simulations are available from 1 December 1980 to 30 November 2080, with RCP8.5 affecting emissions starting 2006, and the timings of passing 2 and 4 °C of warming are also based on 25 year centred running means.

- Figure 4: the use of % as a unit for frequency is very confusing here. If I understand correctly the % is calculated based on number of years (in 25 year climate period) that index values are below a given threshold. This is a guess, it is not clearly explained.
Much later, in Figure 10, the authors use "number of events" instead - a much more straightforward type of unit. I recommend using this unit throughout.

We will clarify "%" and "number of events" to communicate more clearly what they mean and how they are distinct units. % is used for the percent of the time the SI values are below certain thresholds (-1 or -2), while "number of events" is used to count the number of individual drought events with different durations (defined as runs of consecutive negative values).

- LL 199-241: this is a very extensive description of a single figure (see earlier comment). Please reflect critically: what pieces of information are really worth mentioning?
- LL 242-290: same here, figure description is far too lengthy.
We will reduce the descriptions to enhance the most essential information.

- L246: "the fit of the gamma and GEV distributions used in the calculation of SPI and SPEI". So gamma and GEV distributions were fitted apparently..? This should have been explained in the Methods Section!
Thank you for pointing this out, this will be clear when we add the definition and more detailed explanation for SPI and SPEI calculation (see first Methods comment).

- L266: "detrended temperature simulations". Again, please explain his properly in the Methods section – how was the detrending done, for what purpose exactly?
This was explained in the data section on UKCP18-RCM (second to last paragraph):

*While AED increases with rising temperatures, changes in humidity, net radiation and wind speed can also play a significant role. Therefore, we represented AED by PET calculated using Penman-Monteith, which includes the effect of all these variables. This method leads to a more robust correlation between the resulting SPEI and soil moisture under a warming climate compared to using the temperature-only Thornthwaite method (Feng et al., 2017) and is recommended over simpler temperature-based methods (e.g. Dewes et al., 2017), however it is still subject to significant limitations (Milly and Dunne, 2016; Greve et al., 2019). The calculation of PET for the UKCP18-RCM follows the same variant of the Penman-Monteith method used by Robinson et al. (2017), to ensure consistency with CHESS-PE. It uses these variables simulated by the UKCP18-RCM ensemble: specific humidity, pressure at sea level, net downwelling longwave radiation, net downwelling shortwave radiation, wind speed at 10m and daily average surface air temperature. PET was set to zero wherever a calculated value was negative (which occurred for less than 1% of the values overall and, when split by ensemble member and month, also less than 1% for all cases except December in ensemble member 1 with 1.2% of negative values).* **To investigate the influence of the projected temperature trend on changes in SPEI-based droughts and the deviation of SPEI from SPI, we also computed an alternate version of projected SPEI (SPEIdtr−tas) using a detrended version of UKCP18-RCM temperature. For this**, *a linear trend was fitted to, and subsequently subtracted from, the simulated temperature time series for each grid cell and month separately. This detrended temperature dataset was used to compute PET as described above, resulting in a PETdtr−tas variable in which any trend left is due to trends in other variables (specific humidity, radiation, wind speed and pressure) or in interactions between variables. As these variables are closely intertwined in the climate models, this unavoidably introduces a physical discrepancy between temperature and the other variables used in the PET calculation. This is taken into account in the interpretation.*

Same as for the comment on bias adjustment, we will move this from the UKCP18 Data subsection to a dedicated separate Methods subsection, and clearly signpost the explanation of detrending with a subheader in a new subsection on potential evapotranspiration.

L272: "purely temperature-based PET": this seems to suggest that temperature has a strong influence on PET, yet the influence of Radiation is much stronger (linear relationship with PET in Penman equation). Please check the reasoning here, it seems flawed.

We will rephrase this as it is indeed confusing. "Purely temperature-based PET" refers to some quantification methods of PET that rely on only temperature data, e.g. Thornthwaite. In our results, temperature detrending has a large impact because it reduces the saturated vapour pressure, which in combination with unchanged specific humidity leads to lower vapour pressure deficit (relative humidity increases), thus reducing PET.

Figure 6: this is first time Observations are shown in any of the results graphs! Why only now and not in the earlier graphs?

Due to bias correction, RCM-derived reference period statistics for regional averages or single grid cells lie close to the observations, so we decided not to show them in the seasonal cycle and aridity figures, as they would have little added value. In Figure 6, spatial co-occurrence becomes important, which was not explicitly considered in the bias correction, and as such we found it interesting to show observations here.

However, we can see that it could still be good for the reader to see the observations earlier. We will add dots for the observations to the scatter plot in Figure 4, showing their position with respect to the UKCP18-RCM reference period SPI6 and SPEI6. We will also add observations to Figures 2 and 3, which would also help address a comment by Reviewer 1 regarding the comparison of the observations and bias corrected simulations.

Also in Figure 6: a gradual color scale is applied here which makes it impossible to distinguish clearly between the 3 scenarios. Note that in the current representation their seems to be no significant difference between the Reference and +2C scenario.

We will look into amending the colours and/or other line properties to improve readability. The colour scale had the following reasoning behind it: green hues were avoided for the reference period in order to not imply that the '81-'05 reference period is not already affected by climate change. A gradual colour scheme was chosen for intuitive interpretation of the progression from reference period to +2 and +4 °C.

Note: I stopped reading here. Sections 5 and 6 are very lengthy and many of the results point in the same direction. Are all these figures and subsections really needed to make the point stated in the title, that "Projected changes in droughts are strongly influenced by the choice of drought index"?

I strongly recommend that the authors take a critical view of their results and make a selection of the materials that most strongly support their conclusions. Then report these clearly and concisely.

We agree that our results, discussion and key messages can be reported significantly more concisely, and we will reduce the text accordingly. While the point stated in the title is indeed not entirely unexpected, we think the magnitude of the differences between these popular drought indicators may commonly be underappreciated (e.g. in cases where only 1 drought indicator is used to represent overall "drought"), and as such we analysed and demonstrated the differences in great depth across different characteristics of drought.

After a critical re-examination of the text and figures, we will significantly condense the text in Sections 5 and 6 in order to better support the key messages in a concise way, including shorter figure descriptions. With regards to figures: we think all figures have valuable information to add, although we are considering leaving out the overall aridity figure (Figure 2). The figures that show results for all 12 regions (Figures 2, 8, 9, 10 and 11) could convey the essential information when showing results for only 4 regions. We will select 4 regions that best represent the regional variability of the responses, and only keep these for the main text. This will automatically merge Figures 8 and 9 (seasonality) into one, and will allow us to place SPI- and SPEI-based results side by side in a merged version of Figures 10 and

11 (durations), thus effectively decreasing the number of figures in the main text by 2. As some readers might still be interested in all or in specific regions, we will include the results for the other regions in the Supplementary materials.

**References**

Gohar, L., Bernie, D., Good, P., & Lowe, J. A. (2018). UKCP18 derived projections of future climate over the UK. Met Office, Exeter.

Parsons, D. J., Rey, D., Tanguy, M., & Holman, I. P. (2019). Regional variations in the link between drought indices and reported agricultural impacts of drought. Agricultural systems, 173, 119-129.

IPCC, 2021: Annex VII: Glossary [Matthews, J.B.R., V. Möller, R. van Diemen, J.S. Fuglestvedt, V. Masson-Delmotte, C. Méndez, S. Semenov, A. Reisinger (eds.)]. In Climate Change 2021: The Physical Science Basis. Contribution of Working Group I to the Sixth Assessment Report of the Intergovernmental Panel on Climate Change [Masson-Delmotte, V., P. Zhai, A. Pirani, S.L. Connors, C. Péan, S. Berger, N. Caud, Y. Chen, L. Goldfarb, M.I. Gomis, M. Huang, K. Leitzell, E. Lonnoy, J.B.R. Matthews, T.K. Maycock, T. Waterfield, O. Yelekçi, R. Yu, and B. Zhou (eds.)]. Cambridge University Press, Cambridge, United Kingdom and New York, NY, USA, pp. 2215–2256, doi:10.1017/9781009157896.022

---

## Author Response (AR1)

Dear authors,

Your manuscript has now received two external referee reports, to which you have responded in the open discussion. Both referees seem to agree in their assessment that the manuscript is of interest, and can potentially make a valuable contribution to the literature. However they also indicate that the manuscript need to be improved on several aspects. Given the nature of the comments, I classify these as revisions, reflecting the need for considerable changes to the text (for instance the length of the Discussion as indicated by both referees). This also means that I will return a revised manuscript to the referees. You can use your previous replies as a starting point for a revision. Please contact me in case you have any questions. Looking forward to receiving a revised version of your work!

Best regards

Ryan Teuling

**Response**:
Thank you for your decision and for your time handling this manuscript. We have revised it following the reviewer comments and our initial responses, and believe these changes have significantly improved the manuscript. The Discussion text was updated and almost halved in length, and overall the revised manuscript is about 7 pages shorter compared to the initial submission. The Introduction was re-written to better convey the motivation and added value of our study, including explanation of our UK focus. The Methods section has been expanded with the requested additional explanations (e.g. additional detail on bias correction and the SP(E)I calculation).

**In the point-by-point responses below, reviewer comments are included in** black**, author responses in** blue**, and manuscript excerpts in** *italic blue***.**
In some cases, we refer to the relevant manuscript sections instead of including them here, in particular when addressing comments led to substantial modifications throughout entire sections.

There are also some grammar/wording changes not directly addressing reviewer comments, which were not documented below but can be inspected in the tracked changes-document. We also included additional relevant references in the Introduction (justifying the relevance of UK-focussed drought research), Methods (aridity index), Discussion (primarily in 6.3, including work published since the initial manuscript submission), and Conclusions (crediting some of the hydrological modelling work being undertaken using the UKCP18 projections for the UK).

**Response to RC1 (anonymous)**

This paper examines projected changes in drought frequency, extent, seasonality and duration for Great Britain using the latest UKCP18 projections. It analyses the differences between two atmospheric drought indicators (SPI and SPEI) and shows that the choice of drought indicator can have a big impact on the derived drought indicator.

Overall, this is an interesting study that is well written and well presented. The analyses are extensive, well thought out and well executed. I believe the paper will have appeal to a wide readership, however, there are some core issues that need to be resolved to enhance the key messages of the paper. Firstly I think the motivation of the study needs to be more clearly defined and more detail is required on the choice of climate projections and bias correction. Secondly, I strongly encourage the authors to take a critical read of the discussion and shorten it to the core messages – it is currently very long and your interesting results are getting lost.

More detailed comments are provided below that I hope the authors find useful.

Thank you for your time and your review. We believe addressing your comments has substantially improved the quality of the paper. In particular, the core issues on motivation and the choice of climate projections have been addressed in the introduction and methods sections respectively, and the discussion has been drastically shortened to about half its original length.

**Comments**

1. **Motivation.** The motivation for the study needs to be clearer. Currently the introduction reads like a series of definitions rather than a compelling story of why this study should be undertaken. There are two core areas where I think this could be improved:
    1. There have been quite a few studies that have used or compared drought indices (many are cited in your discussion) so what does this study offer that is novel and different?
    2. Why focus on Great Britain (which often isn't thought of as a country that experiences lots of droughts!) and how are your research questions relevant to this region?

We have modified the introduction section throughout to more clearly convey the motivation of the study, including two paragraphs responding to the two core areas mentioned above:

1. Novelty: "*Although previous studies have compared historical and projected changes using these SI in different regions of the world (e.g. Stagge et al., 2017; Chiang et al., 2021), this study adds a new level of detail by an in-depth analysis of different drought characteristics and attention to within-GB regional differences, and is the first to use UKCP18 with these SI to assess projected changes in drought characteristics for GB.*"

2. UK focus: "*This study focuses on Great Britain (GB) to compare projected drought changes based on the SPI and SPEI. Despite not typically being thought of as a particularly drought-prone area, GB has experienced several droughts in the past which lead to widespread impacts, including impacts on ecosystems (including algal blooms and fish kills), agriculture and domestic water supply (Rodda and March, 2011; Kendon et al., 2013; Turner et al., 2018). The impacts of climate change on future droughts in the UK is therefore a key concern for stakeholders including water managers and farmers (e.g. Watts et al., 2015)*"

New references for this paragraph:

*Kendon, M., Marsh, T., and Parry, S.: The 2010–2012 Drought in England and Wales, Weather, 68, 88–95, https://doi.org/10.1002/wea.2101,2013.*

*Rodda, J. and March, T.: The 1975/76 Drought – a Contemporary and Retrospective View, Tech. rep., Centre for Ecology & Hydrology, http://nora.nerc.ac.uk/id/eprint/15011/1/CEH_1975-76_Drought_Report_Rodda_and_Marsh.pdf, 2011.*

*Turner, S., Barker, L. J., Hannaford, J., Muchan, K., Parry, S., and Sefton, C.: The 2018/2019 Drought in the UK: A Hydrological Appraisal, Weather, https://doi.org/10.1002/wea.4003, 2018*

*Watts, G., Battarbee, R. W., Bloomfield, J. P., Crossman, J., Daccache, A., Durance, I., Elliott, J. A., Garner, G., Hannaford, J., Hannah, D. M., Hess, T., Jackson, C. R., Kay, A. L., Kernan, M., Knox, J., Mackay, J., Monteith, D. T., Ormerod, S. J., Rance, J., Stuart, M. E., Wade, A. J., Wade, S. D., Weatherhead, K., Whitehead, P. G., and Wilby, R. L.: Climate Change and Water in the UK - Past Changes and Future Prospects:, Progress in Physical Geography, https://doi.org/10.1177/0309133314542957, 2015*

2. **Use of UKCP18.** There needs to be better justification for the use of the regional projections from UKCP18 in this study – why use this set of projections instead of the local UKCP18 projections for example? Or use a set of climate projections that encompasses different GCMs or RCP scenarios (for example)?

The following paragraphs were added in the Data section to further justify our use of the UKCP18 regional simulations:

*"UKCP18 is the most recent set of national climate projections for the UK and have been produced by the Met Office Hadley Centre with the aim of providing a range of storylines to support adaptation efforts in the UK (Murphy et al., 2018)."*

*…*

*"The ensemble thus does not sample GCM-RCM structural uncertainty, only parameter uncertainty, and was designed to cover a range of possible futures. While multiple GCM-RCM structures would add another interesting dimension to the study, expanding the ensemble was outside the scope and capacity of the study. The horizontal resolution of the RCM simulations is 12km over GB (available on OSGB36 grid projection). As droughts tend to be more spread out in space and time, we judged that the 12km daily resolution of the UKCP18 RCM pose a better trade-off between practicality and spatiotemporal detail than the higher-resolution convective permitting simulations for this study."*

3. **Bias Correction.** At the end of Section 2.2 there is a section on bias correction which needs more detail. Why did you choose these bias correction methods and how are they applicable to the types of biases you observe between the climate projections and observational data? It would be helpful to add some plots in the supplementary information showing these biases to help the reader understand the nature of the biases and how they were corrected. For example, you note in section 3.2 that a direct comparison of the results between climate model ensemble members and observations is only possible because their distributions are similar – it would be helpful to see evidence of this.

The section on bias adjustment was expanded with additional explanation and moved to a dedicated Methods subsection:

*"3.2 Bias adjustment*
*As comparison to observations revealed significant bias in the simulation of both*
*precipitation and PET (see Figs. S1 and S2), these variables were statistically post-*
*processed using the ISIMIP3b change preserving bias adjustment method (Lange, 2019)*
*version 2.4.1 (Lange, 2020). The biases we observed for different quantiles were not equal*
*to the biases observed in the mean, which is why we selected a bias adjustment method that*
*took this into account. Similarly, biases also varied between months and locations, so the*
*bias adjustment needed to be specific for each month and grid cell. The ISIMIP3b bias*
*adjustment method is based on quantile mapping, but also preserves projected changes in*
*the variables being corrected, and enables separate adjustment of the frequency of dry days*
*– a desirable feature for drought research. For precipitation, the gamma distribution and*
*mixed additive/multiplicative per-quantile change preservation were used. For PET and*
*PETdtr−tas, the Weibull distribution, detrending and mixed additive/multiplicative per-*
*quantile change preservation were used. A dry threshold of 0.1 mm day-1 was selected*
*below which there is considered to be no precipitation or PET. In what follows, UKCP18-*
*RCM indicates the bias adjusted data."*

To show the nature of the biases before adjustment, maps of ensemble-averaged
precipitation and potential evapotranspiration bias (in three sections of their distributions)
were included in the supplementary material. As the biases were well-corrected, similar
post-correction bias maps would take up two pages for little added information, and were
thus not included. Instead, to give evidence for the good match between observations and
reference period simulation statistics after bias adjustment, we have added statistics of the
observations to the following plots:

- Fig. 2 and Fig. S3 (seasonal cycles for the different regions)
- Fig. 3 (aridity map)
- Fig. 4 (observations included in scatter plot for two time periods: the 50 year SI
  calibration period and the 25 year reference period)

4. **Discussion.** The discussion section is extremely long and as a result, a lot of your
   interesting results get a little lost amid all the discussion. The authors need to have a
   critical read of the discussion and carefully consider if all the text is needed. As a
   suggestion, I would significantly shorten section 6.4 as this tends to be a review of the
   literature, rather than a discussion of your results with context from the literature (you
   could just use the summary paragraph – you don't really need all the preceding text).

In the revision, the discussion was almost halved in text length, which we think has benefited
readability. For Section 6.4 in particular, we have kept the intro and summary paragraphs
with slight modifications, and removed about 2 pages of text in total. Sections 6.1, 6.2 and
6.3 were also significantly shortened.
Section 6.3 (Role of AED) was largely re-written, to give it more structure for the reader. We
also included some additional relevant literature, including interesting works on evaporation
published since the initial submission of this paper.

**Minor/Technical Corrections**

P2 L45. 'Drought indices that only rely on atmospheric data are a popular choice due to data
availability and **propagating model uncertainties**.' I don't understand this sentence – why
are they a popular choice due to propagating model uncertainties?

*Changed to be clearer as follows: "While indicators exist for variables relevant to different drought types, drought indices that only rely on atmospheric data are a popular choice due to (historical) data availability and due to their ease of use (they do not require the deployment of an impact model, such as a hydrological model)."*

P3 L65. You could add into the third research question the uncertainty in the RCM as a lot of your results focus on the differences between ensemble members. E.g. How sensitive are the projected changes in drought characteristics to the choice of atmosphere-based drought indicator and parametric uncertainty in regional climate models?

*We amended the research question as follows, to include this as well as the GMWL as a source of uncertainty: "How sensitive are the quantified projected changes in drought characteristics to the choice of atmosphere-based drought indicator, and how does it compare to other sources of uncertainty?"*

P3 Section 2.1. It may be useful to state the time-period you used from each observational dataset.

*Added sentence in Section 2.1: "…, using the following time periods: 1961-2010 for the SI calibration (see Section 3.4), 1981-2010 for the bias correction, and 1981-2005 for comparison to the reference period UKCP18-data in this study."*

P5 L141-142. Why did you include aridity as well? What does it add to the story? The motivation could be a little clearer.

*Added to Section 3.4: "While drought refers to a period of below-normal water availability for a given context, aridity refers to the climatic average moisture availability (Dai, 2011). This is included in this study in order to help establish an understanding of the mean climatic changes projected for precipitation and PET in UKCP18-RCM, before proceeding to assessing projected changes in drought characteristics. To this end, the aridity index (AI) was calculated as the annual average ratio of precipitation to PET (e.g. UNEP, 1992; Feng and Fu, 2013; Greve et al., 2019), which is more intuitive to interpret than the standardized indicators."*

P6 L168. The area for London is small, but it is a central hub which contains around 14% of the population of GB and is likely to be significantly affected by droughts in the future. Consequently, leaving out the figures for London because the area is small seems an odd choice, given the significant impacts changes in droughts will have in this small part of GB. Potentially a better reason would be because the results are very similar the South-East region or East of England?

*The results are indeed very similar to those of South-East England, but as we have now reduced the number of figures to be shown in the main text to four, we have also included the results for London in the supplementary materials along with the rest of the regions (Figs. S3-S8).*

Figure 1. It is difficult to see the labels for North and West Scotland – can these be moved or made clearer?

*Label readability has been improved and the selected "main text" regions were highlighted on the map.*

P9 L203-205. This sentence is a little difficult to understand – can it be rewritten?

In the end, this sentence disappeared in the compression of Section 5.1 (see RC2 and responses), because the section now focusses more on the key take-aways from the discussed figures in favour of individually describing the projections for each SI, GMWL and drought category.

P9 L209. 'For extreme meteorological drought, all ensemble members **project multiples** of the reference period frequency by +4 ∘C.' I don't understand what you mean here.

This sentence also disappeared in the compression of Section 5.1.

P10 L233 'due to SPEI6 occurrences beginning to saturate when they have already become quite frequent'. What do you mean by 'beginning to saturate'?

This sentence also disappeared in the compression of Section 5.1.

P28 L666. I think you are missing some key references from the brackets?

This sentence disappeared in the compression of Section 6.4. The references that could have been meant to go here (Touma et al., 2015 or Lee et al., 2019) were included in Section 6.3 instead (L463), for similar reasons as why they would've been cited in the disappeared sentence.

Code and Data Availability – Great to see that the data you produced are available but this section needs to be a full description of all the data used in the study, including links to all the data you used for analysis (i.e. for the CHESS-PE, HadUK and UKCP18 data). I would also reword to 'The SPEI and SPI data **produced in this study** are available on Zenodo (doi:10.5281/zenodo.6123020) (Reyniers et al., 2022b) alongside the bias adjusted UKCP18- based PET (doi:10.5281/zenodo.6320707) (Reyniers et al., 2022a).'

Thanks again for pointing this out, this section now reads:

*"The SPEI and SPI data produced in this study are available on Zenodo (doi:10.5281/zenodo.6123020) (Reyniers et al., 2022b) alongside the bias adjusted UKCP18- based PET (doi:10.5281/zenodo.6320707) (Reyniers et al., 2022a). Python code for the computations and analyses is available upon reasonable request. The CHESS-PE data used in this study was obtained from the UK CEH Environmental Information Data Centre (https://doi.org/10.5285/9116e565-2c0a-455b-9c68-558fdd9179ad) (Robinson et al., 2020). HadUK-Grid data was obtained from the Centre for Environmental Data Analysis (http://dx.doi.org/10.5285/d1343358 (Met Office et al., 2019), as well as the raw UKCP18-RCM simulations (https://catalogue.ceda.ac.uk/uuid/589211abeb844070a95d061c8cc7f6565 (Met Office Hadley Centre, 2018)."*

**Response to RC2 (Marie-Claire ten Veldhuis)**

In this paper changes in drought characteristics are evaluated for GB, for 2 future climate scenarios. Two drought indices are used to characterize drought severity, SPI and SPEI, for various space and time-scales. The study finds increase in most drought characteristics (frequency, extent, duration etc) for future climate conditions, not entirely unexpected.

In particular, the authors emphasize that the choice of drought index influences the quantitative assessments of projected drought changes.

Given this perspective, it is particularly important to document not only the indices used but also the full range of methods applied to reach their conclusions. This is the main problem I see with this paper: many aspects of the methods used for analysis are not clearly explained. Secondly, the Results and Discussion sections are very long and need to be strongly condensed to convey only the essential information. To give an example, section 5.1 covers almost 2 pages to describe a single figure, followed by 2 more pages for a second figure. That's a lot of descriptive information that can be drastically shortened, based on a critical reassessment of what pieces of information are really important and worth for the reader to know.

[concluding reviewer comment moved here as it also addresses the manuscript in general:]

Note: I stopped reading here [after Section 5.1 and Fig. 6]. Sections 5 and 6 are very lengthy and many of the results point in the same direction. Are all these figures and subsections really needed to make the point stated in the title, that "Projected changes in droughts are strongly influenced by the choice of drought index"? I strongly recommend that the authors take a critical view of their results and make a selection of the materials that most strongly support their conclusions. Then report these clearly and concisely.

Thank you for your time and your comments, we believe that the paper has substantially improved after addressing them. The Methods section was expanded and structured to address all comments. The Discussion and Results sections were shortened drastically, both by condensing the text and reducing the volume of Figures shown. Specifically, Section 5.1 was reduced from 92 to 48 lines, and the results and discussion sections were both about halved in text length. The number of figures was reduced by selecting only four regions to show in the main text, whose response is representative of the responses seen across all the analysed GB regions for the seasonal cycle, drought seasonality and duration distribution. These regions were highlighted on the map in Fig. 1, and results for the other regions were shown in the supplementary materials. In total, the paper reduced in length by about 7 pages.

**Detailed comments:**

Abstract:

1. General comment: the summary of results presented here is quite superficial, i.e descriptive rather than interpretive. Deeper interpretation of the results would make the Abstract a lot more appealing.

We re-wrote the abstract to better convey the relevance and context of our results:

"*Abstract. Droughts cause enormous ecological, economical and societal damage, and are already undergoing changes due to anthropogenic climate change. The issue of defining and quantifying droughts has long been a substantial source of uncertainty in understanding*

*observed and projected trends. Atmospheric-based drought indicators, such as the Standardised Precipitation Index (SPI) and the Standardised Precipitation Evapotranspiration Index (SPEI), are often used to quantify drought characteristics and their changes, sometimes as the sole metric representing drought. This study presents a detailed systematic analysis of SPI- and SPEI-based drought projections and their differences for Great Britain, derived from the most recent set of regional climate projections for the UK. We show that the choice of drought indicator has a decisive influence on projected changes in drought frequency, extent, duration and seasonality by 2 ◦C and 4 ◦C above pre-industrial levels. The increases projected in drought frequency and extent are far greater based on the SPEI than based on the SPI. Importantly, compared to droughts of all intensities, isolated extreme droughts are projected to increase far more in frequency and extent, and show more pronounced changes in the distribution of their event durations. Further, projected intensification of the seasonal cycle is reflected in an increasing occurrence of years with (extremely) dry summers combined with wetter than average winters. Increasing summer droughts also form the main contribution to increases in annual droughts, especially using SPEI. These results show that the choice of atmospheric drought index strongly influences the drought characteristics inferred from climate change projections, comparable to the uncertainty from the climate model parameters or the warming level, and therefore potential users of these indices should carefully consider the importance of potential evapotranspiration in their intended context. The stark differences between SPI- and SPEI-based projections highlight the need to better understand the interplay between increasing atmospheric evaporative demand, moisture availability and drought impacts under a changing climate. The region-dependent projected changes in drought characteristics by two warming levels have important implications for adaptation efforts in GB, and further stress the need for rapid mitigation.”*

2. Check phrasing here: the phrasing suggests that projected changes are sensitive to the choice of drought index (L5). However, projected changes are simply what the simulated climate scenarios tell us, how they are summarized in quantitative metrics is where the differences in interpretation come in.

Following our original response to this comment clarifying that our use of projections (i.e., including the quantification step using the SPI and SPEI, which can be considered simple models) follows the IPCC definition, we addressed the issue of potentially confusing uses of "projection" by being clear about which variables are indicated in phrasings such as "projected changes", and by being explicit about the SI being included in the term "projections" by using phrasings such as "quantified using SPI", "SPEI-based projections" or "inferred using SPI". The sentence indicated in this specific comment was not included in the re-written abstract."

3. Same confusing phrasing is used throughout, e.g. (L14) "SPEI results in greater increases in drought frequency and extent". Obviously the drought characteristics do not change, only how the indices are computed.

This specific sentence disappeared in rewriting.

4. L16: "projected changes (..) depend on the drought index, (..)". Again, reasoning is flawed: projected changes are the same, the indices are different, not the other way around.

This specific sentence disappeared in rewriting.

**Introduction**:

- P2, L 39: it is suggested here that evapotranspiration only depends on atmospheric variables, but strictly speaking vegetation also plays a role (stomatal conductance).

This was amended as proposed, but then the sentence was left out in re-writing the introduction.

- P3, l65: same phrasing issue as in Abstract.

Revised version: *"How sensitive are the quantified projected changes in drought characteristics to the choice of atmosphere-based drought indicator, and how does it compare to other sources of uncertainty?"*

Methods:

- P5: it would be helpful to provide the definitions (and/or the equations) of the indices that are used in the paper (AI, SPI, SPEI), so the reader doesn't need to search back in the literature.

For aridity, context and the equation were added to the AI definition paragraph:

*While drought refers to a period of below-normal water availability for a given context, aridity refers to the climatic average moisture availability (Dai, 2011). This is included in this study in order to help establish an understanding of the mean climatic changes projected for precipitation and PET in UKCP18-RCM, before proceeding to assessing projected changes in drought characteristics. To this end, the aridity index (AI) was calculated as the annual average ratio of precipitation to PET (e.g. UNEP, 1992; Feng and Fu, 2013; Greve et al., 2019), which is more intuitive to interpret than the standardized indicators. For time slices of 25 years, this gives*:

$$AI = \frac{1}{25} \sum_{y=1}^{25} \frac{Precipitation_y}{PET_y}$$

For SPI and SPEI, the common general equation was added, and more detail was provided on their calculation so that the reader indeed doesn't need to refer back to the cited literature:

*"The drought indices compared in this study are SPI and SPEI. Both are widely used in the literature to quantify droughts, and they imply contrasting assumptions of the surface water balance: for SPI, no evaporation takes place, while for SPEI, evaporation takes place and is not limited by moisture availability. Multi-scalar standardized climate indicators such as these allow for comparison of unusually dry (or wet) periods across locations with different climates. The SI are calculated as follows. First, the time series of a variable D (precipitation for SPI, precipitation minus PET for SPEI) is aggregated using a specified accumulation period length of n months, such that the value for each month in the resulting time series is the average of that month and the n preceding months. Then, a suitable distribution F D for that variable is fit to the aggregated time series, for each month and location. The SI value for an accumulation period length n at a time step t is then defined as follows:*

$$SIn_t = \phi^{-1}(F_D(D_{n,t}))$$

*with Dn,t indicating D accumulated over the n time steps preceding t (inclusive), and φ the standard normal distribution. Monthly values of SPI and SPEI are calculated using n of 3 to 24 months. Following recommendations provided by Stagge et al. (2015b), the two-parameter gamma distribution was used for calculating SPI and the generalized extreme value (GEV) distribution was used for calculating SPEI. For shorter SPI accumulation periods (1-3 months) and further into the future in the UKCP18-RCM simulations (with drying*

*summers), there may be occurrences of zero accumulated precipitation for grid cells in drier regions. To take this possibility into account, the SPI values corresponding to the probability of zero accumulated precipitation were calculated separately following the method proposed by Stagge et al. (2015b), which avoids the mean SPI becoming larger than 0. A 50-year period (1961-2010) of observation-based data (regridded HadUK-Grid and CHESS-PE) was used to fit the distributions for the SPI and SPEI calculation. This observation-based calibration was also applied to the UKCP18-RCM data to allow a direct comparison of the results between climate model ensemble members and observations. This is appropriate because the bias adjustment brings the distributions of the reference period climate model data close to the observed distributions."*

- P5, L151: "observation-based calibration": this needs clarification. How was this calibration done, this is currently not explained.

  This refers to fitting the distributions to the observation data: *"A 50-year period (1961-2010) of observation-based data (regridded HadUK-Grid and CHESS-PE) was used to fit the distributions for the SPI and SPEI calculation. This observation-based calibration was also applied to the UKCP18-RCM data to allow a direct comparison of the results between climate model ensemble members and observations."*

- P6, section 3.3, Drought characterization: it is stated that spatiotemporal characterization is important - agreed. Unfortunately, the authors do not specify the space and time scales used in their characterization. What is "regional", "seasonal", what range of space and time scales did they investigate?

  This has been clarified by phrasing changes and additions throughout the corresponding paragraphs:

  "*Given the importance of both space (e.g. extent, spatial connectivity, local vulnerability) and time (e.g. seasonal timing, duration) for drought impacts, the spatiotemporal characterisation of droughts is an important element of any drought study. It is approached here in three ways. First, the frequency (fraction of the time in drought) of dry and extremely dry conditions was computed for each individual grid cell of GB separately, for each ensemble member and the observations. Second, the drought area extent was quantified as the fraction of the total GB area simultaneously in (extreme) drought. We then compute the frequency with which different drought extents are exceeded (fraction of time). Third, regionally averaged SI values were used to investigate drought seasonality and duration. For computing these regional averages, we used the UKCP18 administrative regions (ukcp18 data, 2021) shown in Fig. 1, as they represented a decent trade-off between the sizes of the regions, number of regions to compare and relevant differences in climatology, projected changes and societal relevance. For investigating the seasonal contributions to longer-term deficits (seasonality), we compared the 6-month aggregated regionally averaged SI (SI6) for March and September for each year to represent the winter and summer contributions to that year's overall dryness (SI12). Durations of individual drought events are defined as periods of continuously negative regionally averaged SI values reaching a threshold value of -1 or lower, following the theory of runs (Yevjevich, 1967). Each event is then assigned to the time slice (reference period, +2 ∘C or +4 ∘C) that contains its central time step, and the number of occurrences of droughts with different duration categories is assessed. Extreme droughts are identified as events that have a peak (i.e. minimum) SI value below -2.*"

- P6, L162: please clarify definition of 'extreme drought'. At present, the choice of SI<-2 sounds arbitrary.

*This has been elaborated as follows: "In order to compare changes in overall drought conditions to changes in more extremely dry conditions, we consider a category of "all/total drought" covering all SI of -1 and lower, and a category of "extreme" drought covering SI values of -2 and lower. These threshold values are a subset of the classification originally introduced by McKee et al. (1993), which has been extensively used in studies using standardised drought indicators."*

- P6, L177: "a distribution fitted to the relatively short times series". This needs explanation: what distributions were fitted, how exactly?

*This sentence now includes "(see Section 3.4).", which is where the calculation of the SI (including the distribution fitting) is now explained in more detail.*

4. Projected climate changes:

- In the caption of Figure 3 it is mentioned that "after bias adjustment using change preserving quantile mapping" is applied to the ensemble members.
This is not the right place to mention such a data processing step! Please explain adequately in the main text.

*The corresponding Data paragraph has been moved to its own dedicated Methods subsection, and this new bias adjustment subsection has also been expanded with more detail in response to this comment as well as a comment by Reviewer 1.*

5. Projected changes in drought characteristics:
   - L204: the authors refer to "2C above pre-industrial", but as far as I understand their reference scenario is 1981-2005. That's not exactly pre-industrial.. Please clarify or correct.

*The "time slice selection" Methods section has been amended to clarify that the +2 and +4 are relative to pre-industrial levels (1850-1900), not the fixed reference period:*

*"Therefore, to assess the impact of climate change on drought characteristics in scenarios with lower climate sensitivity and more mitigation (resulting in lower warming levels above pre-industrial times), a time slice approach was implemented to investigate changes at two specific global mean warming levels. A common fixed reference period (1981-2005) was used for all ensemble members to compare to these future time slices and observations. For each ensemble member, a time slice was selected from 12 years before to 12 years after the year in which the centred 25-year rolling mean global temperature exceeds + 2 ◦C and + 4 ◦C above pre-industrial levels (defined as 1850-1900) in the driving global model (see Table 2 in Gohar et al. (2018))."*

*The exact sentence this comment referred to disappeared in re-writing the results section.*

- Figure 4: the use of % as a unit for frequency is very confusing here. If I understand correctly the % is calculated based on number of years (in 25 year climate period) that index values are below a given threshold. This is a guess, it is not clearly explained.
Much later, in Figure 10, the authors use "number of events" instead - a much more straightforward type of unit. I recommend using this unit throughout.

*The figure caption now explains this as follows: "Spatially averaged projections of drought frequency, expressed as the fraction of time SI is below the threshold, for each […]."*

*Furthermore, the distinction between the "number of events" unit used for drought duration and the "% of time" unit used in Figure 4 has been clarified in the drought characterisation Methods section.*

- LL 199-241: this is a very extensive description of a single figure (see earlier comment). Please reflect critically: what pieces of information are really worth mentioning?
- LL 242-290: same here, figure description is far too lengthy.

*The section with these figure descriptions (5.1) has been almost halved in length, to better convey the key messages learned from both figures.*

- L246: "the fit of the gamma and GEV distributions used in the calculation of SPI and SPEI". So gamma and GEV distributions were fitted apparently..? This should have been explained in the Methods Section!

*See expanded explanation of SI calculation in the Methods section above.*

- L266: "detrended temperature simulations". Again, please explain his properly in the Methods section – how was the detrending done, for what purpose exactly?

*The corresponding paragraph has been moved from the Data section to its own dedicated Methods subsection.*

L272: "purely temperature-based PET": this seems to suggest that temperature has a strong influence on PET, yet the influence of Radiation is much stronger (linear relationship with PET in Penman equation). Please check the reasoning here, it seems flawed.

*Rephrased to clarify what we meant: "PET calculation methods which only rely on temperature (e.g. Thornthwaite)"*

Figure 6: this is first time Observations are shown in any of the results graphs! Why only now and not in the earlier graphs?

*In the initial submission, they were judged to be more interesting to show in Figures 6 and 7. Observations were now also added to Figures 2, 3 and 4 as well, demonstrating a good match with the reference period simulations for the seasonal cycles of P and PET, aridity index and SPI/SPEI. The latter shows the sensitivity of SPI and SPEI to the period used to calibrate / fit the distributions in their calculation, through the slight difference between the dots showing the 1981-2005 observations and 1961-2010 observations.*

Also in Figure 6: a gradual color scale is applied here which makes it impossible to distinguish clearly between the 3 scenarios. Note that in the current representation their seems to be no significant difference between the Reference and +2C scenario.

*The colour scheme for this figure has been adjusted for enhanced contrast, while still following the same reasoning behind the original colour scale which led to picking a gradual colour scale. Additionally, we re-arranged the order in which the line plots are overlaid, so that the thick lines representing the ensemble means are on top of the thin lines representing individual ensemble members. These changes make it now easier to see the differences between the reference and +2°C scenario for SPI, although the smaller*

magnitude of the differences (compared to SPEI or +4°C) of course also makes them more difficult to distinguish.

All figures in the study were checked for colourblind-friendliness with https://www.color-blindness.com/coblis-color-blindness-simulator/.

---

## Referee Report (RR1)

**Review Comment hess-2022-94**

**Title: Projected changes in droughts and extreme droughts in Great Britain strongly influenced by the choice of drought index**

Thanks to the authors for implementing the extensive revisions, the manuscript has improved a lot. The revised version brings out the most important findings of the study much more clearly. Part of these are primarily of regional interest (changes in drought for study region of Great-Britain, GB) and another part of wider relevance – the sensitivity of drought assessments to the choice of drought index (in this case, precip only or precip and PET, SPI vs SPEI). Especially for the second part, the study has potential to draw additional conclusions to the ones presented so far, that can make the manuscript more attractive and of wider relevance.

Hereby I provide a couple of recommendations to push the study a bit further to enhance its general relevance and interest, for consideration by the editor:

1. The first 2 research questions focus on changes in Precipitation and PET (as expressed in drought indices SPI and SPIE) due to global warming for the region of GB. The authors conclude that droughts increase in all respects (frequency, extent, intensity), with regional differences across GB.
   However, they do not elaborate on the novelties of their study: what are the new conclusions thanks to the "new level of detail by in-depth analysis of different drought characteristics…"
   Similarly, in the Discussion section, the authors state their results "are in broad agreement" with earlier studies without providing much detail on the novelties brought out by the "in-depth analysis".

   This is a missed opportunity: surely, the in-depth analysis brought new insights that are worth discussing (how they go beyond the existing literature) and reporting in the conclusions?

2. This brings up the 2nd point: the authors mention an interesting point of interest for GB as their study area: the fact that "GB sits in the transition between humid, radiation-controlled N-Europe and more arid, precipitation controlled S-Europe."
   So, one expects regions along this transition to be affected differently by global warming in terms of precipitation and especially PET. It's almost imperative to discuss the results in this light: how do the regional differences in drought changes relate to this transition in climatologies? How do the contributions of precipitation and PET to changes in droughts relate to the climatology?

3. The stark differences in drought effects between SPI and SPIE presented in e.g. Fig 5 and Fig 8 clearly suggest that changes in PET have a far stronger contribution to droughts than SPI, especially in SW- and E.-England.
   This raises the question what causes the strong contribution of PET? Is it changes in radiation (global warming is expected to result in more cloudless days in the region), in specific humidity, temperature?
   The authors have only isolated the effect of temperature which seems to be very large, comparing Fig 5-bottom versus Fig5-middle. This is a surprising result, since SPEI is calculated based on specific humidity, so the only remaining temperature effect is caused entirely by its effect on the slope of the CC-curve?. This definitely merits more extensive discussion.
   And it would be very interesting to see how changes in radiation play a role here, especially given that we are looking at a radiation-controlled to moisture-controlled transition.

4. The insight in contributions of climatological variables to changes in PET and thereby changes in SPEI compared to SPI along the climatological transition over GB would be a very interesting conclusion to report and of relevance beyond the regional study of GB.

In summary, I believe the results of this study enable drawing some very interesting additional conclusions and I invite the authors to push a bit further the interpretation of their results to bring these out.

---

## Author Response (AR2)

**Response to Editor Decision**

Dear authors,

Your revised manuscript has been seen by two referees that also provided a review in the online discussion. As you will see, they are generally pleased with the changes made, but also suggest a number of mainly textual changes/additions that would further improve the manuscript. I agree with their assessment that this would improve both the quality and impact of your study. While the changes suggested are generally minor, I have classified the revision as "revision" rather than minor revision because the latter does not allow me to return the manuscript to the reviewers for a final check, and both have indicated to be willing to check a revised version once more. But you can treat it as minor revision.

Best regards, and all the best for the New Year!

**Ryan Teuling**

Dear Ryan,

Thank you for your time and positive decision. Please find below our responses to the individual Reviewer Comments.

RC in **black** Author response in blue Manuscript excerpts in *italic blue*.

Best regards, Nele Reyniers (on behalf of the authors)

**Response to RC 1 (anonymous)**

Thank you to the authors for their detailed revisions of the paper. There have been significant changes to the text and this has made the core messages and novelty of the paper much clearer.

I have a couple of very minor comments, but very happy with the paper overall so please treat these as suggestions.

Thank you for your positive evaluation and for using your time to review our revised manuscript. We implemented all your comments as detailed below.

1. In Figure 2, I can't see the line for the observations. A different colour might be better here? Or perhaps they need their own column of plots?

This was addressed by 1) using a lighter colour for simulated precipitation to improve contrast; 2) using a dashed line for the observations to improve readability, as the observations tend to overlap with the bias-adjusted simulations; 3) emphasizing in the Figure caption that observations are shown in a darker colour.

2. Line 355. "Sustained multi-year droughts are a major concern for water managers (e.g. )." There are no examples in the brackets.

The following missing citation has been added: (Marsh et al., 2007).

3. Line 427-428. "In this section, we discuss the implications of the differences between SPIand SPEI-based projections due to PET increases, and link this to the context of the GB." This sentence is not needed.

The sentence was removed.

4. In Section 6.5 on limitations, you may want to add that the choice of PET dataset to bias correct the climate projections may also have an impact on the final result? There is now a Had-UK-PET dataset available that is quite different to the CHESS-PE data.

Thank you, the following sentence has been added:

Finally, using a different observation-based dataset for bias adjustment of PET such as the recently produced Hydro-PE dataset (Robinson et al., in review), may also lead to quantitative differences in the results.

**References**

Marsh, T., Cole, G., and Wilby, R.: Major Droughts in England and Wales, 1800–2006, Weather, 62, 87–93, https://doi.org/10.1002/wea.67, 2007.

Robinson, E. L., Brown, M. J., Kay, A. L., Lane, R. A., Chapman, R., Bell, V. A., and Blyth, E. M.: Hydro-PE: Gridded Datasets of Historical and Future Penman-Monteith Potential Evaporation for the United Kingdom, Earth System Science Data Discussions, pp. 1–44, https://doi.org/10.5194/essd-2022-288, in review.

**Response to RC 2 (Marie-Claire ten Veldhuis)**

Thanks to the authors for implementing the extensive revisions, the manuscript has improved a lot. The revised version brings out the most important findings of the study much more clearly. Part of these are primarily of regional interest (changes in drought for study region of Great-Britain, GB) and another part of wider relevance – the sensitivity of drought assessments to the choice of drought index (in this case, precip only or precip and PET, SPI vs SPEI). Especially for the second part, the study has potential to draw additional conclusions to the ones presented so far, that can make the manuscript more attractive and of wider relevance.

Hereby I provide a couple of recommendations to push the study a bit further to enhance its general relevance and interest, for consideration by the editor:

**[...]**

In summary, I believe the results of this study enable drawing some very interesting additional conclusions and I invite the authors to push a bit further the interpretation of their results to bring these out.

Thank you for your positive review, the interesting questions raised in your comments and your time spent reviewing our revised manuscript. We have expanded our discussion and conclusions sections based on your comments, while being conscious of not making the discussion too lengthy (as this was a major issue raised by both reviewers in the first round of reviews).

Please find below our responses to your comments. We responded to (1) separately and then grouped (2), (3) and (4) together, because our revisions based on the latter three comments partially overlap, and (4) already tied (2) and (3) together.

1. The first 2 research questions focus on changes in Precipitation and PET (as expressed in drought indices SPI and SPIE) due to global warming for the region of GB. The authors conclude that droughts increase in all respects (frequency, extent, intensity), with regional differences across GB.

However, they do not elaborate on the novelties of their study: what are the new conclusions thanks to the "new level of detail by in-depth analysis of different drought characteristics..." Similarly, in the Discussion section, the authors state their results "are in broad agreement" with earlier studies without providing much detail on the novelties brought out by the "in-depth analysis".

This is a missed opportunity: surely, the in-depth analysis brought new insights that are worth discussing (how they go beyond the existing literature) and reporting in the conclusions?

We addressed this by more clearly highlighting some of the novel findings thanks to our detailed analysis in the discussion and conclusions sections, except for sections where this was already present:

**DISCUSSION:**

**6.1:**

**New added text:**

[revised manuscript text omitted]

- 2. This brings up the 2nd point: the authors mention an interesting point of interest for GB as their study area: the fact that "GB sits in the transition between humid, radiation-controlled N-Europe and more arid, precipitation controlled S-Europe." So, one expects regions along this transition to be affected differently by global warming in terms of precipitation and especially PET. It's almost imperative to discuss the results in this light: how do the regional differences in drought changes relate to this transition in climatologies? How do the contributions of precipitation and PET to changes in droughts relate to the climatology?
- 3. The stark differences in drought effects between SPI and SPIE presented in e.g. Fig 5 and Fig 8 clearly suggest that changes in PET have a far stronger contribution to droughts than SPI, especially in SW- and E.-England. This raises the question what causes the strong contribution of PET? Is it changes in radiation (global warming is expected to result in more cloudless days in the region), in specific humidity, temperature?

The authors have only isolated the effect of temperature which seems to be very large, comparing Fig 5-bottom versus Fig5-middle. This is a surprising result, since SPEI is calculated based on specific humidity, so the only remaining temperature effect is caused entirely by its effect on the slope of the CC-curve?. This definitely merits more extensive discussion.

And it would be very interesting to see how changes in radiation play a role here, especially given that we are looking at a radiation-controlled to moisture-controlled transition.

4. The insight in contributions of climatological variables to changes in PET and thereby changes in SPEI compared to SPI along the climatological transition over GB would be a very interesting conclusion to report and of relevance beyond the regional study of GB.

Thank you for these interesting comments, we are glad our work and discussion sparked more questions. However, comprehensively answering the questions raised would require substantial further analysis far exceeding *minor revisions* and exceeding the scope of our present project (indeed, they would likely deserve a dedicated separate paper). Therefore, we have raised these important points in the discussion and conclusions in the context of the excellent opportunities they represent for further research. For example, for comment 2, it would be interesting to investigate drought changes in the context of the range of evaporation regimes by moving beyond the atmospheric variables that contribute to PET and looking at (simulated) *actual* evapotranspiration and other hydrological variables. However this would require modelling of the surface water balance – something that we are undertaking for a separate paper but only for a restricted region. For comment 3, we did consider analysing all PET variables as a research direction quite early in the study, however we decided it would exceed the scope of the paper.

Rather than undertaking new research to address these interesting questions, we have modified the text to highlight these issues as future avenues. In the results section, we also better clarified the interpretation of the temperature effect in the specific humidity-based formulation of PET we used, and also corrected a wording error we discovered (relative humidity *increases*, not *decreases*, when lowering temperature and retaining specific humidity).

**RESULTS**

[revised existing:] "However, by detrending the temperature, the saturation humidity level computed in the PET calculation was reduced for future simulations, which, combined with the unadjusted specific humidity projections, resulted in artificially increased relative humidity and thus a decreased vapour pressure deficit term."

**DISCUSSION 6.3**

The section focusing on the GB context was expanded to discuss the context of the range of evaporation controls over GB, and the contributions of isolated variables to increases in PET:

[revised existing:] "Interestingly, GB sits in the transition between humid, radiation-controlled Northern and Central Europe and more arid, precipitation-controlled Southern Europe (Teuling et al., 2009). Evaporation is generally more water-limited and negatively correlated with temperature in summer toward the south and east, and more energy-limited and positively correlated with temperature in summer in the north and west of GB on an annual basis (Seneviratne et al., 2006; Kay et al., 2013). This has important implications for the expected impacts of increasing AED on future droughts across GB, as the influence of AED varies between energy- and water-limited evaporation regimes, and the effect of AED increases can be more complicated in transitional regions (Vicente-Serrano et al., 2020). Indeed, ...

[added:] The importance of the range of evaporation regimes for explaining drought propagation and drought impacts across GB has not received much attention in existing literature, but presents a valuable direction for further research. For example, the currently least humid areas of GB are projected to experience large increases in SPEI-drought, increases in aridity, and on average longer and more intense seasons where PET exceeds precipitation. The effect of extreme SPEI-drought conditions on soil moisture and streamflow droughts in these areas could be smaller than suggested by the magnitude of the PET contribution due to moisture availability becoming limited. In such conditions, vegetation may still be significantly impacted due to high AED and its components (Schönbeck et al., 2022). Understanding potential shifts in these evaporation regimes under climate change could help inform climate change adaptation strategies related to land and water use. To better understand the PET component of the projected SPEI-based drought projections for GB, we detrended temperature (which affected the vapour pressure deficit term and the slope of the Clausius-Clapeyron relation), after which no increases in SPEIdtr-tas drought frequency were projected in most regions of GB. Further research into projected changes for the different variables influencing PET (radiation, temperature, relative and specific humidity, wind speed) is needed to better understand the strong contribution of PET to SPEI-based drought projections, and to help understand possible shifts in evaporative regimes over GB."

**CONCLUSIONS**

[added:] "In particular, further research is needed to understand the effects of the contribution of PET to projected drought conditions across the range of climatological evaporation regimes in GB (from energy-limited to transitional and water-limited), and likely changes in these regimes. Moreover, analysing the contributions of changes in radiation, relative and specific humidity, temperature and wind speed can shed light on the PET component itself."